# Sparling: End-to-End Spatial Concept Learning via Extremely Sparse Activations

**Kavi Gupta**
Department of Electrical Engineering and Computer Science
Massachusetts Institute of Technology
Cambridge, MA 02139, USA
`kavig@mit.edu`

**Osbert Bastani**
Department of Computer and Information Science
University of Pennsylvania
Philadelphia, PA 19104, USA
`obastani@seas.upenn.edu`

**Armando Solar-Lezama**
Department of Electrical Engineering and Computer Science
Massachusetts Institute of Technology
Cambridge, MA 02139, USA
`asolar@csail.mit.edu`

## Abstract

Real-world processes often contain intermediate state that can be modeled as an extremely sparse activation tensor. In this work, we analyze the identifiability of such sparse and local latent intermediate variables, which we call *motifs*. We prove our Motif Identifiability Theorem, stating that under certain assumptions it is possible to precisely identify these motifs exclusively by reducing end-to-end error. Notably, we do not assume identifiability of parameters, but rather of a latent intermediate representation output by a local model, thus allowing these representations to be arbitrarily complex functions of the input. Additionally, we provide the Sparling algorithm, which uses a new kind of informational bottleneck that enforces levels of activation sparsity unachievable using other techniques. We confirm empirically that extreme sparsity is necessary to achieve good intermediate state modeling. On synthetic domains, we are able to precisely localize the intermediate states up to feature permutation with $> 90\%$ accuracy, even though we only train end-to-end.

## 1 Introduction

A hallmark of deep learning is its ability to learn useful intermediate representations of data from end-to-end supervision via backpropagation. However, these representations are often opaque—values in the intermediate vectors generally do not map to meaningful concepts. As a consequence, there has been a great deal of recent interest in concept bottleneck models (Koh et al., 2020), which guide models towards meaningful concepts at intermediate layers. Training these models either relies on supervision of the intermediate concepts or on designing algorithms capable of learning these from end-to-end supervision. The latter is desirable since supervision of concepts is only possible in domains where they are known *a priori*, yet a key advantage of deep learning is the capability of learning representations beyond handcrafted knowledge. However, learning concepts end-to-end is a daunting task—there is a huge space of possible concepts that could produce the same labeled input/output mapping.

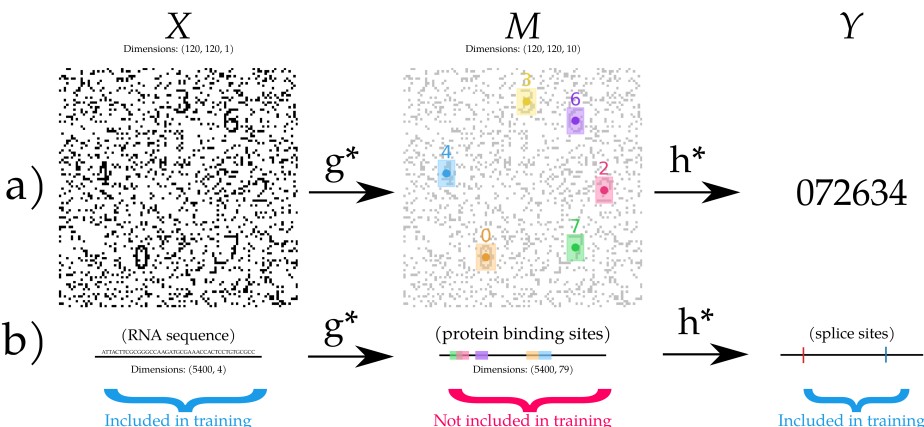

Figure 1: (a) Example of the DIGITCIRCLE domain, alongside (b) a cartoon of the splicing problem. The input $x$ is mapped by the ground truth $g^*$ function to the motif map $m^*$ of the positions of every digit/protein binding sites, which is itself mapped by the ground truth $h^*$ function to the output $y^*$, the sequence 072634/splice sites. Only $x$ and $y^*$ are available during training; the goal is to reconstruct $g^*$ and $h^*$. Note that in splicing, unlike DIGITCIRCLE, the motifs can overlap. The $M$ dots indicate the representation as described in Section 2, which is a one-hot encoding at each location.

A recent work in genomics (Gupta et al., 2024) has demonstrated a capability similar to the one described above: it learns "motifs" (intermediate spatial concepts) from an end-to-end training signal. In that work, the motifs represented locations in RNA where proteins bind, which can be assumed to be sparse, and for which there are some approximate models. The paper did extensive validation to demonstrate that end-to-end training led to more accurate motif models. This result was surprising because even though no additional direct information about the motifs was provided, the prediction of these motifs became more aligned with independent experimental measurements. Inspired by this result, we analyze some theoretical questions raised by this approach: mainly under what conditions is it possible to faithfully recover an intermediate signal. We find that we can dispense with the approximate starting motif models altogether by making stronger assumptions about the domain.

In this work we advance a theorem demonstrating that, assuming that a true process contains intermediate concepts which are sparse, local, and necessary and sufficient to produce the output, it is possible to learn them end-to-end (up to simple transformations) without intermediate supervision. Notably, apart from the model before the sparse intermediate layer being local, we do not impose any constraints on model architecture[1].

To learn the "correct" spatial concepts, or motifs, we need to impose significant structure on the intermediate representation. As an example, consider the DIGITCIRCLE task shown in the top of Figure 1. In this task, the input is a noisy image of a circle of digits, and the label is a list of the digits read counterclockwise starting from the smallest. Then, a motif is a digit in the input image $x$, and the intermediate representation (which we call the *motif space*) consists of a binary indicator $m[i, j, c] \in \{0, 1\}$ for each position $(i, j)$ and digit $c$ indicating whether $c$ occurs at $(i, j)$ in $x$. These kinds of problems also arise in other domains—for another example (illustrated at the bottom of Figure 1), predicting where RNA is spliced is a key problem in genomics, and splice sites can be predicted based on which proteins bind to the RNA sequence; binding sites can be determined based on local RNA sequence motifs (Gupta et al., 2024). In the context of these examples, we wish to characterize a set of assumptions under which it is statistically possible to learn motifs from end-to-end supervision alone. To demonstrate that these assumptions are satisfiable, we advance an algorithm that is capable of identifying motifs from end-to-end supervision.

---

[1]As such, we do not claim identifiability of parameters, but rather of the $\hat{g}$ function, as described by input/output behavior.

Our key insight is that spatial concepts typically have two key properties: (i) *locality*—i.e., a motif $m[i, j, c]$ only depends on the image[2] in a window around the corresponding spatial position $x[i, j]$, and (ii) *sparsity*—i.e., since there are far fewer spatial concepts than pixels, only a tiny fraction of motif activations are nonzero. We prove a theoretical result that locality and sparsity, together with several reasonable assumptions about the distribution of training examples, suffices to recover the motif space from a statistical standpoint. This result requires that we can find the true optimizer of the loss; thus, we additionally provide an algorithm for optimizing end-to-end error in a sparse model, and demonstrate its efficacy on three datasets.

**Contributions.** We present three main contributions in this paper. First, we provide a proof of our Motif Identifiability Theorem: that sparse local latent variables are identifiable. We attempt to make as few assumptions as possible about the structure of the relationships between the inputs, motifs, and output, assuming only that the motif patches are separated and are both necessary and sufficient to computing the output. We do not make any further assumptions regarding the structure of the functions relating the motifs and the output (e.g., limited number of layers). Second, we describe the SPARLING algorithm, which allows for training models with an extreme sparsity constraint. We accomplish this via a layer that sets activations below some threshold equal to zero; this threshold is iteratively updated to achieve a target sparsity level (e.g., 99%). In order to address the unstable optimization landscape this produces at high sparsity values, our optimization algorithm anneals the target sparsity over time. Finally, we demonstrate several domains in which SPARLING can correctly identify the intermediate latent variable. These domains, while synthetic, demonstrate that the identifiability guarantee we proved is achievable in practice.

**Related work.** We summarize the related work here, and provide a more detailed discussion in Appendix A. Most existing work on learning interpretable latent representations assume some prior knowledge about the representations, including both concept bottleneck models and the Genomics work mentioned above. The recently proposed "Language in a Bottle" technique (Yang et al., 2023) proposes to address this problem by using large language models (LLMs) to identify intermediate concepts; however, this is only applicable to certain domains. Our theoretical work is connected to the statistical literature on identifiability, which asks whether the "true" parameters of a model can be recovered from data. Indeed, prior work has proposed algorithms that guarantee identification of latent variable models such as Hidden Markov Models (HMMs) (Yoon, 2009) and Probabilistic Context-Free Grammars (PCFGs) (Hsu et al., 2012). While the problem is similar, we are interested in the deep learning setting where the latent concepts form the intermediate layer between two arbitrary models (presumably neural networks). Then, our theoretical results establish assumptions on the models and data distributions under which we can guarantee recovery of the "true function". This problem is similar to that of nonlinear Independent Component Analysis (ICA) (Hyvärinen et al., 2023; Khemakhem et al., 2020), where the goal is identifying independent components mixed by some nonlinear function. However, we attempt to make much more limited assumptions of the "mixing function" and show that small end-to-end error is sufficient to imply recovery of the latent concepts. While our algorithm is not guaranteed to achieve small end-to-end error, this is a useful theorem as verifying low end-to-end error is trivial given a test set. Additionally, we find that in our experiments we do achieve low end-to-end error.

## 2 PRELIMINARIES

We are interested in settings where intermediate activations represent latent variables corresponding to semantically meaningful concepts in the problem. To this end, we consider the case where the *ground truth* is represented as a function $f^* : X \to Y$ composed $f^* = h^* \circ g^*$ of two functions $g^* : X \to M$ and $h^* : M \to Y$. We call the latent space $M$ the *motif* space.

We consider the task of training $\hat{g}$ and $\hat{h}$ to accurately model $g^*$ and $h^*$ using only end-to-end data $\mathcal{D} = \{(x, f^*(x)) : x \sim \mathcal{D}_X\}$ (i.e., enforcing only that their composition $\hat{f} = \hat{h} \circ \hat{g}$ accurately models $f^*$)[3]. Importantly, we assume no access to data on $M$ (in particular, which components of

---

[2]We use the term "image" and assume 2 dimensions in our running examples, but our results apply to any tensor input, which could be a 1D sequence, 3D, etc.

[3]We do not consider noise for the purposes of this paper. The result could be modified to handle IID Bernoulli noise in the error function by replacing the end-to-end error with end-to-end error minus irreducible error in the theorem statement.

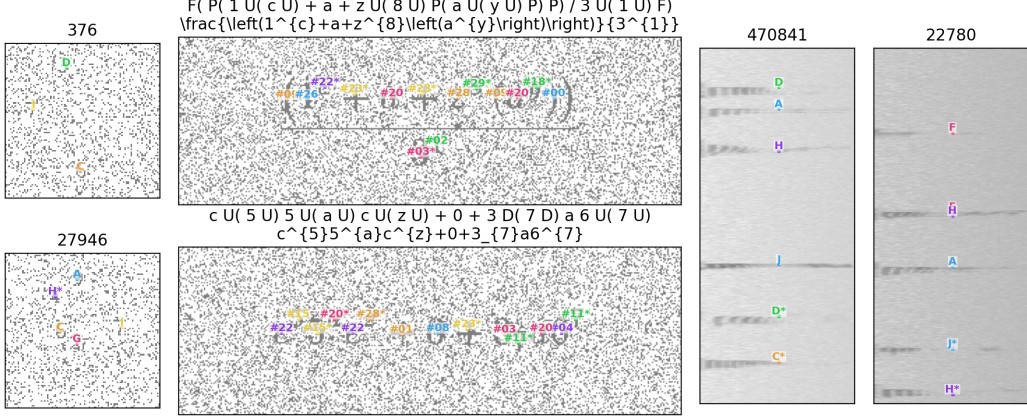

Figure 2: Two examples of inputs (images), outputs (sequences in titles), and our $\hat{g}$ predictions for seed=1 (colored dots) for DIGITCIRCLE, LaTeX-OCR, and AUDIOMNISTSEQUENCE. For LaTeX-OCR, we provide the output twice, first as the sequence of commands generated by the network and second as the translation of those commands into LaTeX. We place a dot for every maximal motif, colored/labeled by the channel that it appears in (e.g., the 0th channel is A or #00, 1st is B or #01, etc.). Stars indicate sites where non-maximal motifs are present as well.

$M$ are active for any particular input). Our goal is to establish the conditions under which this task is possible and to present an algorithm to derive $\hat{g}$ and $\hat{h}$. Specifically, we focus on the case where $g^*$ and $\hat{g}$ exhibit the properties of locality and sparsity as described below.

We assume that elements of $X \subseteq \mathbb{R}^{I \times [d]}$ and $M \subseteq \{0, 1\}^{I \times [n]}$ are tensors, where $[d] = \{1, ..., d\}$, and where $I = [D_1] \times \ldots \times [D_l]$ is a set of spatial indices, $d$ is the number of input channels, and $n$ is the number of kinds of motif. In addition, $Y$ is a discrete label space. $M$ acts as a one-hot encoding for each motif, as depicted in Figure 2, with the last channel corresponding to which motif is which. In the formulation for our theorem, $M$ is assumed to be binary, but of course during training, $M$ can be treated as a real-valued tensor; an additional step should be added to discretize it to binary for the purposes of applying the theorem.

As a running example, consider the DIGITCIRCLE task in Figure 1, where the input $x \in X$ is a monochrome image and the label $y \in Y$ is a sequence of digits in that image from left to right. In this example, $I = D_1 \times D_2$ is the height and width of the image, and $d = 1$ for the monochrome channel. The motif embedding $m \in M$ has $n = 10$ motif kinds, and encodes the occurrences of digits—specifically, $m[i, j, c]$ encodes whether digit $c$ occurs at position $(i, j)$. Then, $g^*$ predicts whether a digit occurs at each position in the input image, and $h^*$ extracts the digit sequence by reading off values of $c$ counterclockwise starting from the smallest digit).

In our example, it is easy to see that $g^*$ satisfies the key properties of *locality* (i.e., its prediction $m[i, j, c]$ depends on the input image $x$ in a window around $(i, j)$) and *sparsity* (i.e., most of the motif values are $m[i, j, c] = 0$). We describe these two properties in more detail below.

**Locality** We define the set $\mathcal{G}$ of "local" motif models to capture convolutional models. SPARLING relies on locality to treat different parts of the input as independent, alongside the MOTIF-SUFFICIENCY assumption we introduce later.

Formally, we define the set $\mathcal{G}$ relative to some convolutional radius $r \in \mathbb{N}$. We then say $g \in \mathcal{G}$ corresponds to a local version $g_l \in \mathbb{R}^{(2r+1) \times \ldots \times (2r+1)} \to \{0, 1\}^n$ such that $g(x)[i, c] = g_l(x[p(i)])[c]$, where $p(i) = \{i - r \ldots i + r\} \times \ldots \times \{i - r \ldots i + r\}$ is the local set of indices. We also define the "motif cell" $p_2(i)$ as being twice as wide $\{i - 2r \ldots i + 2r\} \times \ldots \times \{i - 2r \ldots i + 2r\}$; corresponding to all locations whose motif footprints overlap the motif at $i$.[4] $|p_2| = |p_2(i)|$ is a

---

[4]Note: while we define this concept relative to a convolution, it can be generalized to graph convolutions or other local operations, just with different definitions of $p$ and $p_2$

constant in $i$ and appears in our error bound. In the case of DIGITCIRCLE, we have that $r = 8$ so $p(i) = \{i - 8 \ldots i + 8\}^2$ and $p_2(i) = \{i - 16 \ldots i + 16\}^2$.

**Sparsity** Let the number of motifs for a channel $c$ in a given activation pattern $m = g(x)$ be $\#_c(m) = \sum_{i \in I} \mathbf{1}(m[i, c] \neq 0)$. We can then define the mean value of this over the dataset for a given motif function as $\#_c(g) = \mathbb{E}_x[\#_c(g(x))]$. Let $\#(t) = \sum_c \#_c(t)$ for both elements of $M$ and $\mathcal{G}$. Let the *density* of a model be $\delta(g) = \#(g)/|I \times [n]|$ and $\delta^* = \delta(g^*)$. We refer to $1 - \delta(g)$ as the *sparsity* of $g$. In general, in our experiments $\delta^*$ tends to be extremely low: for example, since we use an average of 4.5 digits for the DIGITCIRCLE domain, and the images are $100 \times 100$, and there are 10 kinds of motif, so $\delta^* = 4.5 \times 10^{-5}$ fraction of motifs are nonzero.

# 3 MOTIF IDENTIFIABILITY THEOREM

The claim that one can reconstruct an intermediate variable from end-to-end data sounds impossible, as there are several ways that such an intermediate variable can be non-unique. In this section, we establish a small set of conditions under which such an intermediate variable might be non-unique and demonstrate that if none of these conditions apply, motifs can be identified.

## 3.1 THEOREM STATEMENT

We define *Motif Identifiability* as a property of a data distribution $\mathcal{D}_X$ and mechanism $g^*, h^*$. Intuitively, it says that for any estimate $\hat{f} = \hat{h} \circ \hat{g}$ of $f^*$, if $\hat{f}$ has low end-to-end error, then $\hat{g}$ must have low motif error (i.e., $\hat{g}$ is a good estimate of $g^*$). In other words, if we are able to learn a model on $(x, y^*)$ data that achieves good end-to-end error, then we can conclude that we have correctly estimated $m^*$ even if we do not have any data on $m^*$. Formally, in Section 3.3 we define three properties: NON-OVERLAPPING, MOTIF-SUFFICIENCY, and $\alpha$-MOTIF-NECESSITY, such that if all these properties hold, then for some $k = O\left(\frac{\#_{\max}^2 |p_2| n^2}{\#^* \alpha^2}\right)$, we have

$$\forall \hat{g} \in \mathcal{G}, \hat{h} \ . \ \delta(\hat{g}) = \delta^* \implies \left(\forall \epsilon > 0, \mathcal{E}(\hat{h} \circ \hat{g}) < \epsilon \implies \mathcal{E}_m(\hat{g}) < k\epsilon\right)$$

where $\mathcal{E}$ is end-to-end error and $\mathcal{E}_m$ is motif error, as defined in Section 3.2. Note that $\delta(\hat{g}) = \delta^*$ enforces extreme sparsity, as $\delta^*$ is must be extremely small due to NON-OVERLAPPING, and thus corresponds to an extreme sparsity setting; in practice $\delta^*$ is found via the adaptive sparsity algorithm, as described in Section 4.

For simplicity, we describe our error metrics and assumptions as if $n = 1$, that is, there is only one kind of motif. We provide multi-kind versions of these formally in Appendix C.

## 3.2 ERROR METRICS

We define error metrics for both end-to-end error and motif error in two ways: a mathematically simple definition for our proof, and a more intuitive definition for our empirical findings (see Section 5.1). We demonstrate that these are equivalent modulo a constant factor in Appendix F. For our proofs, we define end-to-end error as exact match: $\mathcal{E}(\hat{f}) = \mathbb{E}_{x, y^* \sim \mathcal{D}}[\hat{f}(x) \neq y^*]$.

Defining the motif error metric, $\mathcal{E}_m$, is more complex. In particular, the definition of equivalence needs to account for $\hat{g}$ placing the motifs at slightly different locations, or permuting the motif channels. Thus, we only check that the predicted point be within the motif cell of a given true motif. In this section, we assume there is only one channel, so there is no channel permutation problem, but in Appendix C.1, we handle channel permutations by taking a minimum over all possibilities.

For our proofs, we define motif error using an intersection-over-union-inspired metric. For the "intersection" in this metric we use the number of true motif cells in $g^*(x)$ covered by a unique motif in $\hat{g}(x)$. To define this, we first define the function $v_{\hat{m}}(i)$ to be the number of motifs in the motif cell surrounding $i$ in $\hat{m} = \hat{g}(x)$:

$$v_{\hat{m}}(i) = \sum_{i' \in p_2(i)} \mathbf{1}(\hat{m}[i'] \neq 0)$$

We then define $u(\hat{g}(x), g^*(x))$ to be the number of motif cells in the true motif pattern $g^*(x)$ that are covered by exactly one motif in the predicted motif pattern $\hat{g}(x)$.

$$u(\hat{m}, m^*) = \sum_{i \in I} \mathbf{1}\left(m^*[i] \neq 0 \wedge v_{\hat{m}}(i) = 1\right)$$

In Figure 2, we represent these with circles (stars correspond to ones with more than one match).

We then take the expectation of $u$ over the dataset to get our "intersection" value. For our "union" value, we take the maximum of the expected number of motifs produced by $g^*$ and $\hat{g}$: $\max(\#(\hat{g}), \#(g^*))$. The result is our metric

$$\mathcal{E}_m(\hat{g}) = 1 - \frac{\mathbb{E}_{x \sim \mathcal{D}}\left[\sum_{c'} u(\hat{g}(x), g^*(x))\right]}{\max(\#(\hat{g}), \#(g^*))}$$

This metric is directionally correct under all circumstances, rewarding $\hat{g}$ that produce motifs that overlap cells of $g^*$ with a lower error[5].

## 3.3 Assumptions

We have three main assumptions, under which our theorem applies. Formal versions of these assumptions can be found in Appendix B.

**NON-OVERLAPPING** This assumption states that motifs cannot appear too near each other; specifically that any two motifs' $p_2(i)$ cannot overlap. This assumption, as written, technically excludes some of our domains because of how large the distance needs to be between two motifs for this condition to be met. This assumption is kept strong for simplicity, and can probably be weakened in future work if an additional assumption stating that a motifs' pattern must use portions of the entire $p(i)$ cell is added. [6]

**MOTIF-SUFFICIENCY** We wish to ensure that the motifs are *sufficient* to predict the output, that is, there is no information necessary to predict the output not captured by the locations of the motifs. To accomplish this, we assert that the specific pixels representing motifs must be independent of $P_m(m)$, the overall structure of the input (i.e., the spatial positioning of each motif relative to the overall image and other motifs). In the context of DIGITCIRCLE, this assumption corresponds to the fact that the patterns for each digit are generated independently of the procedure which decides which digit goes where. There is an additional assumption that the background (the part of the image that does not correspond to motifs) must be translation invariant, a property that is satisfied by independent-and-identically distributed noise, but also by selections of clips from a larger object.

This is our main assumption, which pairs with sparsity and locality to mean that motifs are independent entities rather than simply correlated subfeatures of some larger picture.

$\alpha$-**MOTIF-NECESSITY** This assumption states that no kind of motif is entirely ignored (or treated as interchangeable with another kind) by the true $h^*$ model. The assumption is designed to be very weak, allowing for a variety of cases in which individual motifs being perturbed does not alter the output; but does require that in cases whose probabilities sum to $\geq \alpha$, a motif pattern $m_1$ must be perturbable by deleting or altering a single motif into a plausible pattern $m_2$ such that these have different outputs $h^*(m_1) \neq h^*(m_2)$.

This assumption is generally easy to satisfy with $\alpha > 0.1$ in domains like DIGITCIRCLE or AUDIOMNISTSEQUENCE where perturbations that alter the output are common and objects with more degrees of freedom (e.g., more digits) have correspondingly lower probability. See Appendix B.3 for a practical example of how to bound $\alpha$.

---

[5]If we assume NON-OVERLAPPING we also have $\mathcal{E}(g^*) = 0$

[6]At present, we assume no overlap at all between $p_2(i)$ cells because we must allow the possibility that a motif is perfectly predictable from, e.g., one pixel in the top left corner of $p(i)$, which is, in nearly every realistic case, impossible. We leave formalizing a more complex version of the assumption that excludes this possibility while allowing overlapping cells to future work.

### 3.4 PROOF SKETCH

We give a proof of this theorem in Appendix E. In short, we proceed by contrapositive, assuming high motif error. We then establish via a counting argument that since $\delta(\hat{g}) = \delta^*$, any motif error must either be due to false negatives or confusion (a channel of $\hat{g}$ being used for two different motifs). In both cases, we then establish that this error must apply to some fraction of all motif sites (via MOTIF-SUFFICIENCY), then establish that this should lead to a perturbation described in $\alpha$-MOTIF-NECESSITY with some proportional probability, and thus to end-to-end error.

## 4 METHODS

SPARLING trains models with *Spatial Sparsity Layer*s using the *Adaptive Sparsity Algorithm*. A python implementation of SPARLING is available in the `sparling` pypi package.

**Spatial Sparsity Layer** This layer is the last step in the computation of $\hat{g}$ and enforces its sparsity. We define a spatial sparsity layer to be a layer with a parameter $t$ whose forward pass is computed

$$\text{Sparse}_t(z) = \text{ReLU}(z - t)$$

Importantly, $t$ is treated as a constant in backpropagation and is thus not updated by gradient descent. Instead, we update $t$ using an exponential moving average of the quantiles of batches[7]:

$$t_n = \mu t_{n-1} + (1 - \mu)q(z_n, 1 - \delta),$$

where $t_n$ is the value of $t$ on the $n$th iteration, $z_n$ is the nth batch of inputs to this layer, $\mu$ is the momentum (we use $\mu = 0.9$), $\delta$ is the target density, and $q : \mathbb{R}^{B \times d_1 \times \ldots \times d_k \times n} \times \mathbb{R} \rightarrow \mathbb{R}^n$ is the standard `torch.quantile` function. $q$ is applied across all dimensions except the last: it produces a value for each channel that represents the threshold $u$ for which the proportion of elements above $u$ in the tensor at that channel is $\delta$. We describe an alternative in Appendix K.3. Since $t_n$ is fit to the data distribution, we can treat this as a layer that enforces that $\hat{g}$ has a sparsity of $1 - \delta$. Finally, we always include an affine batch normalization before this layer to increase training stability. We provide an analysis on the necessity of this addition in Appendix K.2.

---

**Algorithm 1** Train Loop $(\hat{f}, \mathcal{D}, M, B, d_T, \delta_{\text{update}})$

---

$T_0 \leftarrow 1$
**for** $t = 1$ **to** $\ldots$ **do**
    TRAINSTEP$(\hat{f}, \mathcal{D}_{Bt:B(t+1)})$
    $T_t \leftarrow T_{t-1} - Bd_T$
    **if** $Bt \mod M = 0$ **then**
        $A_t \leftarrow \text{VALIDATE}(\hat{f})$
        **if** $A_t > T_t$ **then**
            $(\hat{f}.\delta, T_t) \leftarrow (\hat{f}.\delta \times \delta_{\text{update}}, A_t)$

---

**Adaptive Sparsity Algorithm** We found that applying an extreme sparsity requirement (very low $\delta$) upon initial training of the network leads to the network getting stuck in a local minimum due to a lack of learning signal. To resolve this, we use a technique inspired by simulated annealing and reduce $\delta$ slowly over time. Annealing hyperparameters is a known technique (Sønderby et al., 2016), but we tie this annealing to end-to-end validation accuracy in order to be flexible to training schedule. As shown in Algorithm 1, we add a step to our training loop that checks validation accuracy $A_t$ and reduces the density whenever it exceeds a target $T_t$, reducing $T_t$ over time. Our experiments use evaluation frequency $M = 2 \times 10^5$, batch size $B = 10$, $d_T = 10^{-7}$, and $\delta_{\text{update}} = 0.75$.

## 5 EXPERIMENTS

### 5.1 EXPERIMENTAL SETUP

We describe our three new domains below. See Figure 2 for examples of each domain.

---

[7]For numerical stability, we accumulate batches such that $|z_n|\delta \geq 10C$ before running this update

**DIGITCIRCLE domain.** The input $x$ is a $100 \times 100$ monochrome image with 3-6 unique digits placed in a rough circular pattern, with some noise being applied to the image both before and after the numbers are placed. The output $y^*$ is the sequence of digits in counterclockwise order, starting with the smallest number. The latent motifs layer $m^*$ is the position of each digit: which can be represented as a $100 \times 100 \times 10$ tensor with 3-6 nonzero entries. Note that we have no access during training and validation to the concept of a digit as an image, nor to the concept of a digit's position.

**LATEX-OCR domain.** As a more realistic test, we take inspiration from Deng et al. (2016) and present the task of synthesizing LATEX code from images. This task is an OCR task like DIGIT-CIRCLE, but with variation in digit rendering (size, aliasing) and a more complex $h^*$.

**AUDIOMNISTSEQUENCE domain.** In this domain, we synthesize short clips of audio representing sequences of 5-10 digits over a bed of noise. The task is to predict the sequence of characters spoken. Here, we test if motif models can generalize: we train and validate with AUDIOMNIST (Becker et al., 2018) samples from Speakers 1-51 and test with samples from Speakers 52-60.

**Splicing domain.** We also considered the splicing domain discussed in Gupta et al. (2024). Since it does not satisfy our assumptions from Section 3.3, SPARLING is not able to precisely identify the motifs, but does perform substantially better than random chance. See Appendix M for our results.

**Architecture and training.** Our neural architecture is adapted from that of Deng et al. (2016). For DIGITCIRCLE, we make $\hat{g}$ have a $17 \times 17$ overall window, by layering four residual units (He et al., 2016), each containing two $3 \times 3$ convolutional layers. We then map to a 10-channel bottleneck where our Spatial Sparsity layer is placed. Our $\hat{h}$ architecture is a max pooling, followed by a similar architecture to Deng et al. (2016). We keep the LSTM row-encoder, but replace the attention decoder with a column-based positional encoding followed by a Transformer (Vaswani et al., 2017) whose encoder and decoder have 8 heads and 6 layers. Throughout, except in the bottleneck layer, we use a width of 512 for all units. For LATEX-OCR we use the same architecture but with 32 motifs (to account for the additional characters) and a $65 \times 65$ overall window (to account for the larger characters, though we find $33 \times 33$ does not change the results substantially). For AUDIOMNISTSEQUENCE we process the audio via a spectrogram with a sample rate of 8000 and 64 channels, use a 33-wide 1D resnet stack for $\hat{g}$ and a transformer for $\hat{h}$. We generate training, validation, and test sets randomly. For efficiency, LATEX-OCR is looped on $10^7$ training samples, the rest are infinite. We use a batch size of 10 and a learning rate of $10^{-5}$. Our validation and test sets both contain $10^4$ examples. Details on computational usage are in Appendix N.2.

**Error Metrics** For our empirical analysis, we use more granular error metrics. First, we define end-to-end error as normalized edit distance:

$$\text{E2EE}(\hat{f}) = \mathbb{E}_{x,y^* \sim \mathcal{D}} \left[ \frac{\text{EDITDISTANCE}(y^*, \hat{f}(x))}{\max(|y^*|, |\hat{f}(x)|)} \right].$$

Next, we disaggregate motif error's false positives, false negatives, and mis-identified motifs into three separate metrics: False Positive (FPE), False Negative (FNE), and Confusion Error (CE) (confusion error occurs when multiple motif channels are confused, this is always zero if $n = 1$). Appendix D.2 contains formal definitions of these metrics and Appendix F contains a proof that these metrics are bounded within a constant multiplicative factor of $\mathcal{E}_m$.

## 5.2 RESULTS

**Motif error.** We show our three metrics of motif error in Figure 3 for each of our models on each domain. Motif errors for our model average below 10% for all our domains, except in the case of FNE on LATEX-OCR. The generally low motif errors, despite only training and validating end-to-end, demonstrate that our algorithm achieves Motif Identifiability on all three domains. This property even holds when generalizing to unseen samples in the AUDIOMNISTSEQUENCE experiment, providing evidence that SPARLING is genuinely learning the motif features rather than memorizing. The one case where our model has high error, FNE on LATEX-OCR, demonstrates the importance of the $\alpha$-MOTIF-NECESSITY assumption: recognizing LATEX text in the space we generated does not require identification of fraction bars or all of ( ) +. For more details, see Figure 2 and Appendix H. Interestingly, this only affects the unimportant digits; this is because our proof is still mostly valid if some motifs are never used: they can simply be treated as part of the background instead.

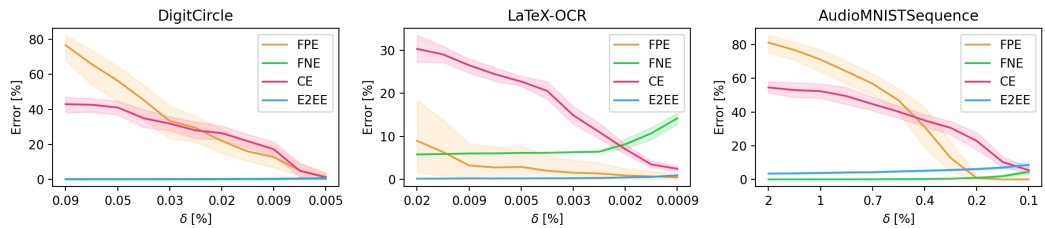

Figure 3: Motif Error, across three different metrics. Bar height depicts the mean across 9 seeds, individual dots represent seed, the error bar represents a 95% bootstrap CI. AUDIOMNISTSEQUENCE has an FPE of exactly 0. High FNE on LATEX-OCR is due to fraction bars, parentheses, and plus signs not being recognized in all cases since it is possible to infer the output without access to these. For a comparison of our technique to less-sparse models, see Figure 4.

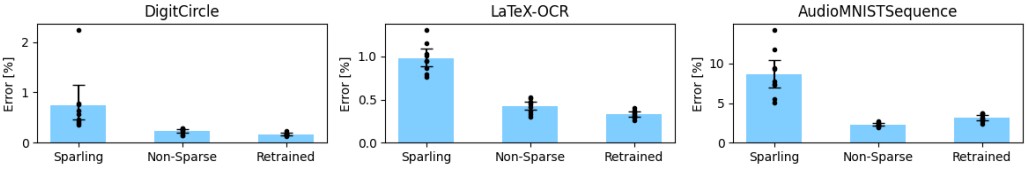

Figure 4: Motif and end-to-end error metrics versus $\delta$. Note that the $x$ axis is a reversed log-scale, since the adaptive sparsity algorithm starts with high density and narrows it exponentially.

**Examples.** Figure 2 shows a few examples for one of our models' intermediate layers. As can be seen, all digits are appropriately identified by our intermediate layer, with very few dots (in these examples, none) falling away from a digit. Note that the activations are consistent from sample to sample—for example, in DIGITCIRCLE, motif C is used for digit 6 in both images.

**Necessity of Extreme Sparsity** Figure 4 shows our error metrics plotted against the sparsity, with the $x$-axis reversed to show progression in training time as we anneal $\delta$. As expected, as $\delta$ decreases, FPE decreases and FNE increases. More interestingly, we note a trade-off between E2EE and CE: as $\delta$ decreases, E2EE increases and CE decreases substantially. This demonstrates a trade-off between a more accurate overall model, which benefits from greater information present and a more accurate motif model, which benefits from a tighter entropy bound. Furthermore, CE is often substantially higher for even a 2-3× increase in $\delta$, demonstrating the need for extreme sparsity. This validates the Motif Identification Theorem, which relies on $\delta(\hat{g}) = \delta^*$ to make its guarantees.

**End-to-End error** As seen in Figure 5, SPARLING tends to produce higher end-to-end errors than a baseline Non-Sparse model. We hypothesize that this is because our constraint on the information flow requires the model to "commit" to a choice on whether or not a given site is a true motif. To

Figure 5: *Retrained* tends to perform as well as or slightly worse than *Non-Sparse*, making up most of the gap from SPARLING. The apparent improvement from *Non-Sparse* to *Retrained* should not be interpreted as real, the numerical difference is tiny and the sample accuracies overlap.

test our hypothesis, we investigate the *Retrained* setting, in which we remove the bottleneck, freeze the motif model $\hat{g}$, and finetune $\hat{h}$ on the training set until convergence. The Retrained setting tends to perform similarly to the Non-Sparse setting, demonstrating that the motif model itself provides signal sufficient to learn the end-to-end function, and validating our hypothesis.

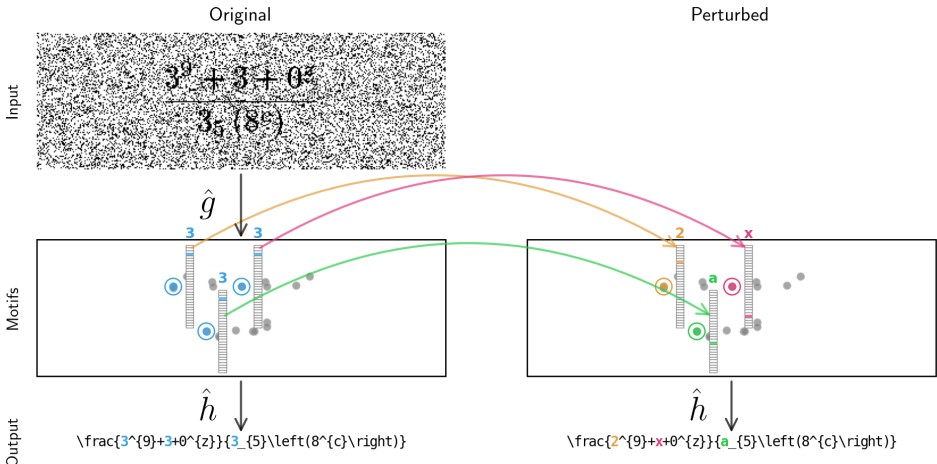

Figure 6: Illustration of $\hat{h}$ model via perturbations of the motif layer. Left column is the original, unmodified flow, with the 3 motifs annotated. On the right, we have the flow when perturbing these motifs; altering each site to 2, a, and x respectively. We then run $\hat{h}$ on the altered motif layer, and note that each of the corresponding positions in the output has been modified.

**Sensitivity of $\hat{h}$ model to perturbations** To demonstrate that the $\hat{h}$ model is behaving as expected, we look at the motifs layer of an example from LATEX-OCR and manipulate it so that certain motifs are changed to others. We then run $\hat{h}$ and ensure that the output is altered as expected. Figure 6 depicts a version of this experiment. As a more quantified notion of how "accurate" $\hat{h}$ is, we perform the following experiment: take 1000 dataset examples where $\hat{f}$ correctly predicts the output (this is done for simplicity so perturbations can be considered with respect to the true output), then take all motifs of a given type (e.g., 3 in the example in Figure 6) and perturb them to some other value.[8] Then we check to see if the output changed as expected, using exact match accuracy.[9] We observe accuracies of 99.3% on DIGITCIRCLE, 86.1% on LATEX-OCR and 93.4% on AUDIOMNISTSEQUENCE, demonstrating that $\hat{h}$ is mostly behaving as expected[10].

## 6 CONCLUSION

We prove that Motif Identification is solvable under certain assumptions. Additionally, we demonstrate SPARLING, a practical algorithm to learn end-to-end models that have a sparse intermediate layer. Finally, we demonstrate that Motif Identifiability is not solely theoretical: SPARLING achieves interpretable and accurate motifs with zero direct supervision on the motifs across three domains.

---

[8]We take care to ensure the resulting motif pattern remains in-distribution: in DIGITCIRCLE we never change to a digit that already exists (since all training examples exclusively contain unique digits), in LATEX-OCR we always start and end with a digit or letter.

[9]For LATEX-OCR and AUDIOMNISTSEQUENCE this is a find and replace; for DIGITCIRCLE we may also need to reorder the output to canonicalize it.

[10]LATEX-OCR probably has higher failure rates because the "wrong subchannel" is being selected, since in the LaTeX domain we allow multiple channels to map to each motif. this means for some characters the small and large forms are being identified distinctly

ACKNOWLEDGMENTS

We thank Christopher B. Burge, Phillip A. Sharp, Chenxi Yang, and Kayla McCue for helpful discussions. This work was supported by NSF grant numbers CCF-1918839 (to A. S-L.) and CCF-1917852 (to O.B.).

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

# A    ADDITIONAL RELATED WORK

**Learning RNA/DNA motifs** In Gupta et al. (2024) the authors introduce the concept of Sparse Adjusted Motifs. Specifically, they model the problem of splicing as a two stage process, in which first proteins bind the RNA sequence, and then cause the sequence to be spliced at certain points. Using end-to-end data of a sequence annotated with splicepoints, as well as baseline models of protein binding patterns in RNA, they are able to improve these models of protein binding. To accomplish this they use the baseline model to predict protein binding affinity, then apply SPARLING, a sparse layer, with a sparsity of $1 - 2\delta$. They then modify this with a neural network trained residually, allowing it to only influence nonzero sites, then apply another SPARLING layer with sparsity $1 - \delta$. In this work, we eschew the complexity off the Adjusted Motif model and instead consider the sparse layer by itself. In Tseng et al. (2024) and Liao et al. (2022), the authors learn motifs without intermediate supervision, but in these cases they heavily restrict the model class of the motif models, requiring them to be 1-layer convolutions.

**Concept bottleneck models.** Previous work also learns models with intermediate features that correspond to known variables. Some techniques, such as Concept Bottleneck Models (Koh et al., 2020) and Concept Embedding Models (Zarlenga et al., 2022), involve additional supervision with existing feature labels. Other techniques, such as Cross-Model Scene Networks (Aytar et al., 2017), use multiple datasets with the same intermediate representation. The Language in a Bottle technique (Yang et al., 2023) uses LLMs to identify intermediate concepts; however this is only applicable to certain domains (e.g., asking an LLM to produce the protein binding motifs in an RNA sequence will result in it providing a list of motif finding tools, not motifs). In this work, we do not require the presence of additional datasets or annotations.

**Identifiability** The problem of identifiability, in which the behavior of some component of a function is inferred via the behavior of the overall function, under some assumptions, is typically set up as an attempt to infer the values of specific parameters up to some isomorphism. In Hsu et al. (2012), the parameters are those of a PCFG expressing a distribution over sequences and the behavior of the function is the computation of a moment of this distribution (with infinite data). In Bona-Pellissier et al. (2023), the parameters are those of a multi layer ReLU network, identifiability is established with infinite data under several assumptions relating to the network as a piecewise linear function. Other work such as Zhong et al. (2017) focuses on strong convexity guarantees on the neighborhood of the true parameters, which is a far stronger claim as it leads to plausible inference algorithms; though the model class is restricted to 1 layer neural networks. In Ahuja et al. (2022), the result of sparse perturbations (perturbations of only some variables) to the latent variables is given, which enables identifiability; this differs from our $\alpha$-MOTIF-NECESSITY in that we only assume the existence of perturbations that affect the observable output, rather than requiring access to these modified outputs as part of the dataset. In our case, we are attempting to infer the motif function $\hat{g}$ rather than any particular parameter, also up to isomorphism. As a result, we make weaker architectural assumptions about $\hat{g}$ and $\hat{h}$. However, the property we attempt to establish is stronger than identifiability with infinite data: we wish to show that the error in identification of $\hat{g}$ is bounded by a multiple of the end-to-end error. While this does not immediately lead to an inference algorithm, it implies that any inference algorithm that preserves our sparsity constraint while achieving low error will be a valid algorithm for identifying the true $g^*$.

**Neural Input Attribution.** SPARLING is useful for identifying the relevant parts of an input. One existing technique that accomplishes this goal is saliency mapping (Simonyan et al., 2013; Selvaraju et al., 2016), which uses gradient techniques to find which parts of the input affect the output most. Another technique, analyzing the attention weights of an attention layer (Mnih et al., 2014), only works with a single layer of attention and does not necessarily produce valid or complete explanations (Serrano & Smith, 2019). Additionally, Amortized Explanation Techniques produce a subset of features that form a "local explanation," i.e., features sufficient to produce a prediction (Jethani et al., 2021). The main benefit a sparse annotation provides over these techniques is unconditional independence: when using sparsity, you have the ability to make the claim "region $x[r]$ of the input is not relevant to the output prediction, regardless of the rest of the input $x[\bar{r}]$". This is a direct result of sparsity and locality and is unavailable when using saliency or attention techniques which inherently condition on the values you provide for $x[\bar{r}]$. Techniques such as Sparse Explanation Values (Sun et al., 2024) do not have this guarantee, and so while they apply to a wider variety of

model structures, they can thus only reason about local perturbations, providing local explanations of changes in behavior.

**Disentangled representations.** *Disentangled representations* are ones where different components of the representation encode independent attributes of the underlying data (Desjardins et al., 2012; Higgins et al., 2016). Locatello et al. (2019) suggests there are no universal solutions to this problem, and all attempts require some prior about the kinds of representations being disentangled. We focus here on a prior regarding sparsity and locality.

**Informational bottleneck.** Other work also constrains the information content of the intermediate representation in a neural network. Strategies include constraining the dimension of the representation—e.g., PCA and autoencoders with low-dimensional representations (Bourlard & Kamp, 1988), or adding noise—e.g., variational autoencoders (Kingma & Welling, 2014). However, these approaches often encourage entangling features to communicate them through a smaller number of channels, and as such do not always learn interpretable representations of an intermediate state.

**Sparse activations.** Note that this notion of sparsity differs from *sparse parameters* (Tibshirani, 1996; Scardapane et al., 2017; Frankle & Carbin, 2018; Ma et al., 2019; Lemhadri et al., 2021; Lachapelle et al., 2023), *sparse causal graphs* (Moran et al., 2021; Lachapelle et al., 2022; Enouen & Liu, 2022; Ren et al., 2024), and *sparse jacobians* (Zheng et al., 2022; Brady et al., 2023); instead this line of work attempts to constrain the information content of an intermediate representation by encouraging *sparse activations*—i.e., each component of the representation is zero for most inputs. Sparse parameters serve different objectives and require different strategies to be used effectively. As sparse parameters only provide interpretability for single or two-layer models, they are generally used for efficiency rather than interpretability in larger models. In terms of imposing sparsity, different techniques must again be used as sparse activation patterns depend on the input, so occasional pruning—e.g., Frankle & Carbin (2018)—is insufficient. Strategies for achieving sparse activations include imposing an $L_1$ penalty on the representation or a penalty on the KL divergence between the representation's distribution and a low-probability Bernoulli distribution (Jiang et al., 2015). However, these techniques typically only achieve 50%-90% sparsity, whereas SPARLING can achieve $> 99.9\%$. We directly compare with these in Appendix K.1. Additionally, Bizopoulos & Koutsouris (2020) uses a quantile-based activation limit equivalent to both of our ablations (see Appendix K.2) combined, but in the simpler context of linear $\hat{h}$ and $\hat{g}$ models. Similarly, Xu et al. (2024) provides an identifiability result given sparse activations, but in the context of an affine model, whereas we allow arbitrary nonlinearity.

## B   FORMAL ASSUMPTIONS

We assume our data generation process is represented by a graphical model $x \leftarrow m^* \rightarrow y^*$; intuitively, the motifs $m^*$ are sampled first, and then $x$ and $y^*$ are sampled conditioned on $m^*$. This allows us to describe our assumptions as constraints on $P(x|m^*)$ and $P(y^*|m^*)$.

There are several ways for this graphical model to be non-unique given the joint distribution $p(x^*, y^*)$; we identify a set of assumptions that excludes any possible such non-uniqueness. To ensure the pixels (or audio samples, nodes, etc.) corresponding to each motif are not just a part of some larger image, we assume motifs cannot appear near each other (NON-OVERLAPPING) and $P(x|m^*)$ must be easily decomposed into factors in order to constrain the relationship between $x$ and $m^*$, ensuring that $x$ is a product of distributions describing the footprints of motifs (MOTIF-SUFFICIENCY). This is our main assumption, analogous to a Markovian assumption in a Hidden Markov Model. Next, we exclude the possibility that some motifs are always redundant: $\alpha$-MOTIF-NECESSITY describes the relationship between $m^*$ and $y^*$, asserting that all motifs are important in some cases; in other words, $h^*$ cannot systematically ignore any motif or treat any two motifs as interchangeable. This assumption ensures the definition of "motif" is restricted to concepts that are possible to learn from end-to-end data, analogous to a full-rank covariance assumption in Linear Regression.

While these constraints may appear strict, they fit problems where $g^*$ identifies small local patterns in the input—e.g. motifs such as the individual digits in DIGITCIRCLE—that are all used at least

sometimes by $h^*$. However, they do *not* fit the splicing domain (primarily NON-OVERLAPPING and MOTIF-SUFFICIENCY), necessitating the additional data used by Gupta et al. (2024).

## B.1 NON-OVERLAPPING

We assume that motif cells cannot overlap in samples drawn from $\mathcal{D}_X$

$$\forall x \in X, P_x(x) > 0 \implies \forall i, i' \in g^*(x), i \neq i' \implies p_2(i) \cap p_2(i') = \emptyset$$

where $i \in g^*(x)$ if $(g^*(x))[i] \neq 0$.

## B.2 MOTIF-SUFFICIENCY

We assert that probability $P_x(x)$ decomposes to independent distributions for each patch $p(i)$ for which $g^*(x)[i] \neq 0$ and a background probability covering all non-patch inputs. Formally, we define the probability of $x$ given that it produces the motif pattern $m$ as

$$P[x|g^*(x) = m] = \left( \prod_{i \in m} P_f(x[p(i)]) \right) P_b\left(x[r(m)]\right)$$

where $P_f$ is a distribution over "foreground" parts of the input (those containing motifs) and $P_b(x)$ is a distribution over "background" $x$, and we are taking a marginal $x[r(m)]$, where $r(m) = I \times [d] \setminus \bigcup_{i \in m} p(i)$ is the set of all indices not in any motif footprint. We then represent $P[x] = \sum_{m \in M} P[x|g^*(x) = m] P_m(m)$ where $P_m(m)$ is our distribution over $m$.

We also require that $P_b$ be translationally invariant. Specifically, for all sets $L \subseteq I$ and all offsets $o \in \mathbb{Z}^l$ such that $\{i + o : i \in L\} \subseteq I$, we have $P_b(x[L]) = P_b(x[\{i + o : i \in L\}])$. That is, the joint distribution should be the same at each location regardless of translation. This property holds for all datasets created by clipping random components of larger datasets, e.g., clipping sequences of RNA from the genome or snippets of text from a book. See Appendix G.1 for a motivating counterexample.

## B.3 $\alpha$-MOTIF-NECESSITY

We assert that the motifs are important, that is, all of them are "used" to compute the output. Being "used" implies two properties. First, we assert that perturbations to motifs must cause $h^*$ to produce different outputs. However, this is insufficient, because it is possible for a perturbation to cause $h^*$ to produce a different output, but for this perturbation to be "correctible" by some alternate $\hat{h}$. To exclude this possibility, we add our second property, that our distribution is structured in such a manner that all perturbations result in a motif pattern that remains in-distribution. These two conditions are sufficient, but they are too strong to apply to most problems of interest, so we relax them by requiring that they co-occur in only some fraction $\alpha$ of motif patterns. Our assumption is parameterized by this $\alpha$; motif importance with a higher $\alpha$ implies that motif errors will lead to end-to-end errors on a higher fraction of the input.

We begin by defining a *perturbation function* $R(m_1)$ that relates a motif $m_1$ to a set $m_2 \in R(m_1)$ which corresponds to $m_1$ with a motif deleted. We then define $R'(m_1) = \{m_2 \in R(m_1) : h^*(m_1) \neq h^*(m_2)\}$; that is the subset of $R(m_1)$ such that the output is changed. The idea here is that the model not accurately reporting a given motif means that it cannot distinguish the conditions that lead to $m_1$ and $m_2$ from each other, but these have different outputs, so it must be incorrect in at least one of these cases.

Ideally, we might want $R'$ to map every $m_1$ to some unique $m_2$ with identical probability. If this is the case, we can guarantee that the model is incorrect either on $m_1$ or $m_2$ and therefore is incorrect in 50% of cases. However, in practice, this is far too strong an assumption. A given motif pattern might have no motifs to delete, or multiple possible deletions, and additionally, many $m_1$ might lead to the same $m_2$ post perturbation.

As such, we define $\alpha$-MOTIF-NECESSITY as an assumption over the following bipartite flow problem: let nodes $\{A_m\}_{m \in M}$ and $\{B_m\}_{m \in M}$, where $A_m$ has input flow $P_m(m)$ and $B_m$ has output flow $P_m(m)$. Add unconstrained connections between $A_{m_1}$ and $B_{m_2}$ if and only if $m_2 \in R'(m_1)$. Our assumption is that the max flow on this graph is at least $\alpha$.

As an intuition, the flow on this graph represents how much weight we put on each perturbation, allowing us to ensure that we always start and end at a high-probability section of the distribution, with the output constraints preventing us from "overusing" any particular $m_2$.

In practice, for both DIGITCIRCLE and AUDIOMNISTSEQUENCE, we can prove $\alpha$-MOTIF-NECESSITY for high $\alpha$ (above 0.25). This works because while several possible deletions on different $m_1$ can lead to the same $m_2$, the $m_1$s each have more degrees of freedom and thus lower probability.

Consider a simplified version of DIGITCIRCLE where there are 3-6 identical digits randomly placed in a line rather than a circle, with 10 possible positions but no two digits can be directly adjacent. We have for an $m_1$ with 5 digits

$$P(m_1) = \frac{1}{4} \qquad\qquad [P(5\text{ digits})]$$

$$\times \frac{1}{21} \qquad\qquad [\text{counted via enumeration}]$$

so $P(m_1) = \frac{1}{84}$ whereas for a $m_2$ with 4 digits

$$P(m_1) = \frac{1}{4} \qquad\qquad [P(4\text{ digits})]$$

$$\times \frac{1}{70} \qquad\qquad [\text{counted via enumeration}]$$

so $P(m_1) = \frac{1}{280}$, and thus $P(m_1) = \frac{3}{10}P(m_2)$. We end up being unable to send the full flow, with each $m_2$ corresponding to between 1 to 3 $m_1$ elements, and in fact are only able to send 0.196 units of flow, out of the available 0.25. Repeating this for $m_1$ with 4 and 6 digits (we cannot delete digits when starting with 3), we end up with an overall $\alpha = 0.465$.

## C    MULTIPLE MOTIF KINDS

We have to modify several definitions to handle the case of multiple channels. However, none of these changes modify the fundamental character of the theorem. Our provided proof (Appendix E) is for the more general case.

One useful definition is that of the set of true motifs: we define $\omega_c(m)$ to be a set of indices corresponding to motif of channel $c$: $\omega_c(m) = \{i \in I : \exists c, m[i, c] \neq 0\}$, we have that $i \in m \iff \exists c, i \in \omega_c(m)$.

### C.1    MOTIF ERROR

Since there are multiple channels, and there is no way for $\hat{g}$ to know *a priori* what the appropriate assignment of motif to channel is, the predicted motifs models should be deemed equivalent to the ground truth model —which is known when we test—if there exists a channel assignment for which they are equivalent.

We follow a similar metric to the one described in Section 3.2 except channel-specific and then minimized over all assignments $\tau : [n] \to [n]$ of channels of $\hat{g}$ to channels in $g^*$:[11].

Our definition of $v$ is modified by identifying a channel

$$v_{\hat{m}}(i, c') = \sum_{i' \in p_2(i)} \mathbf{1}(\hat{m}[i, c'] \neq 0)$$

___________________________________

[11]We do not require $\tau$ to be a permutation, as in practice we might want to allow extra channels in case we do not know the exact number. In this case, the metric will only provide a good score if a real $g^*$ motif is split up among channels of $\hat{g}$, but not if a single channel $c'$ of $\hat{g}$ corresponds to multiple channels of $g^*$, which would be a loss of information

We then modify $u$ to be the number of motif cells of channel $c$ in the true motif pattern $g^*(x)$ that are uniquely covered by a motif of channel $c'$.

$$u(\hat{m}, m^*, c', c) = \sum_{i \in I} \mathbf{1}\left(m^*[i, c] \neq 0 \wedge v_{\hat{m}}(i, c') = 1 \wedge \forall c'' \neq c', v_{\hat{m}}(i, c'') = 0\right)$$

We then take the sum of this over all channels $c'$ of $\hat{g}$ and corresponding channels $\tau(c')$ of $g^*$, and then take an expectation over the dataset to get our "intersection" value, the expected number of true motif cells covered by a unique predicted motif of the corresponding channel.

$$\mathbb{E}_{x \sim \mathcal{D}}\left[\sum_{c'} u(\hat{g}(x), g^*(x), c', \tau(c'))\right]$$

Our union value is unchanged, leading to the metric:

$$\mathcal{E}_m(\hat{g}) = \min_\tau \left(1 - \frac{\mathbb{E}_{x \sim \mathcal{D}}\left[\sum_{c'} u(\hat{g}(x), g^*(x), c', \tau(c'))\right]}{\max(\#(\hat{g}), \#^*)}\right)$$

### C.2 MOTIF-SUFFICIENCY ASSUMPTION

We slightly modify our assumption to

$$P[x|g^*(x) = m] = \left(\prod_{c \in [n], i \in \omega_c(m)} P_c(x[p(i)])\right) P_b\left(x[r(m)]\right)$$

which is identical except that $P_f$ is replaced by a $P_c$, which is distinct for each channel.

### C.3 $\alpha$-MOTIF-NECESSITY ASSUMPTION

Adding more channels means we need to consider multiple perturbation functions. Specifically, we consider two classes of *perturbation functions* $R(m_1)$ that relate pairs of motif patterns, those where $m_2 \in R(m_1)$ correspond to $m_1$ with motif of a particular channel $c_1$ deleted, and those where $m_2 \in R(m_1)$ corresponds to $m_1$ with a particular motif of channel $c_1$ mutated into a motif of channel $c_2$.

One additional subtlety is that we also require flexibility to shifts in the perturbed motif's position, ensuring that the precise positions of motifs are not determined by the rest of the motifs, precluding a situation where, e.g., motifs are aligned to a grid, and the learned $\hat{g}$ "sneaks through" information about a motif's channel via off-grid positioning. Note that this means $\alpha$-MOTIF-NECESSITY is tied to locality and in particular only makes sense when $g^*$ is local within $p_2$ as defined in Section 2.

Let $\Delta = \{-2r \ldots 2r\} \times \ldots \times \{-2r \ldots 2r\}$ be the constant set such that $p_2(i) = i + \Delta$.

Our formal definition then is over $\mathcal{R}$, a set of perturbation relations defined as follows:

- For all $d_1 \in \Delta$ and $c_1$ let there be some $R \in \mathcal{R}$ such that $m_2 \in R(m_1)$ if and only if there exists some $i \in I$ such that $m_1$ and $m_2$ agree everywhere except that $m_1[i + d_1][c_1] \neq 0$ and $m_2[p_2(i)] = 0$
- For all $d_1, d_2 \in \Delta$ and $c_1 \neq c_2$ let there be some $R \in \mathcal{R}$ such that $m_2 \in R(m_1)$ if and only if there exists some $i \in I$ such that $m_1$ and $m_2$ agree everywhere except that $m_1[i + d_1][c_1] \neq 0$ and $m_2[i + d_2][c_2] \neq 0$

Then for each $R \in \mathcal{R}$ we assert the existence of some maximum flow of $\alpha$ on the corresponding $R'$.

For the proofs, we conceptualize this flow as a probability distribution $\psi(m_2|m_1)$ which corresponds to the flow from $m_1$ to $m_2$ as a fraction of $P(m_1)$, with $\psi(\perp|m_1)$ taking the remainder of the flow. Since it corresponds to this flow, it must satisfy the properties

$$\sum_{m_1 \in M} \psi(m_2|m_1) P_m(m_1) \leq P_m(m_2)$$

and

$$\sum_{m_1 \in M, m_2 \in M} \psi(m_2|m_1) P_m(m_1) \geq \alpha$$

# D EVALUATION METRIC DETAILS

## D.1 PRELIMINARIES

We now define our FPM and MM motif sets, along with the $C$ function.

**Predicted motifs.** For a given predicted motif tensor $\hat{m}$, we define $P(\hat{m}) = \{(i, c') : \hat{m}[i, c'] > 0\}$ to be the set of motifs predicted in $\hat{m}$, where $i \in I, c \in [n]$. Typically, we are interested in the set of motifs $P(\hat{g}(x))$ for our estimated motif model $\hat{g}$.

**Footprint identification.** Let $C : I \times M \to I \cup \{\perp\}$ be a function that identifies the motif cell that a given index is within, or $\perp$ otherwise:

$$C(i', m^*) = i \iff i' \in p_2(i)$$

By NON-OVERLAPPING, this is always unique, but we can extend the definition to be coherent otherwise by giving it flexibility to choose an arbitrary such $i$:

$$(C(i', m^*) = i \impliedby i' \in p_2(i)) \wedge (C(i', m^*) = \perp \iff \forall i, i' \notin p_2(i))$$

**False Positive Motifs.** We now have the ability to define our first class of motifs: *false positive motifs*. These are predicted motifs that do not correspond to any real motifs:

$$\text{FPM}(\hat{m}, m^*) = \{(i', c') \in P(\hat{m}) : C(i', m^*) = \perp\}.$$

We denote the remaining motifs by

$$P_1(\hat{m}, m^*) = P(\hat{m}) \setminus \text{FPM}(\hat{m}, m^*).$$

**Maximal Motifs** First, we need to define the set of all predicted motifs that cover the same footprint as a given predicted motif. We do so via the $A_{\hat{m}, m^*}$ function, which takes a given predicted motif (assumed to overlap some footprint) and returns all others covering the same footprint:

$$A_{\hat{m}, m^*}(i', c) = \{(i'', c') \in P(\hat{m}) : C(i'', m^*) = C(i', m^*)\}$$

Now we can define *maximal motifs* are predicted motifs that are maximal in the footprint they cover:

$$\text{MM}(\hat{m}, m^*) = \{t \in P_1(\hat{m}, m^*) : \hat{m}[t] = \max_{t' \in A_{\hat{m}, m^*}(t)} \hat{m}[t']\}$$

We can also define *non-maximal motifs* are predicted motifs that are non-maximal in the footprint they cover:

$$\text{NMM}(\hat{m}, m^*) = \{t \in P_1(\hat{m}, m^*) : \hat{m}[t] \neq \max_{t' \in A_{\hat{m}, m^*}(t)} \hat{m}[t']\}$$

However, we ignore non-maximal motifs entirely for the purposes of our analysis, under the reasoning that these are trivially removable in practice.

## D.2 MOTIF ERROR METRIC

We then define three motif error metrics that we use empirically in evaluating our learned $\hat{g}$ models.

First, the *false positive error (FPE)* is the percentage of motifs that are false positive motifs.

$$\text{FPE}_{\mathcal{D}}(\hat{g}) = \frac{\mathbb{E}_{x \sim \mathcal{D}}[|\text{FPM}(\hat{g}(x), g^*(x))|]}{\mathbb{E}_{x \sim \mathcal{D}}[|P(\hat{g}(x))|]}.$$

Second, the *false negative error (FNE)* is the percentage of true sites that are not covered by any motif.

$$\text{FNE}_{\mathcal{D}}(\hat{g}) = \frac{\mathbb{E}_{x \sim \mathcal{D}}[|\{(i, c) \in P(g^*(x)) : \nexists(i', c') \in P(\hat{g}(x)) : i' \in p_2(i)\}|]}{\mathbb{E}_{x \sim \mathcal{D}}[|P(g^*(x))|]}.$$

Finally, the *confusion error (CE)* is defined as follows: (i) rearrange $\hat{g}$'s channels to best align them with $g^*$, (ii) compute the percentage of maximal motifs in footprint of a true motif that do not correspond to the true motif's channel:

$$\text{CE}_{\mathcal{D}}(\hat{g}) = \min_{\tau:[n]\to[n]} \frac{\mathbb{E}_{x \sim \mathcal{D}}[|\text{conf}_\tau(\hat{g}(x), g^*(x))|]}{\mathbb{E}_{x \sim \mathcal{D}}[|\text{MM}(\hat{g}(x), g^*(x))|]},$$

$\text{conf}_\tau(\hat{m}, m^*)$ represents the motifs that do not match ground truth under rearrangement $\tau$

$$\text{conf}_\tau(\hat{m}, m^*) = \{t \in \text{MM}(\hat{m}, m^*) : \neg\text{mat}_\tau(t, C(t, m^*))\}|$$

and $\text{mat}_\tau(t, t^*)$ is a function that checks whether the two motif index tuples match under channel rearrangement $\tau$.

A low FPE/FNE implies that the model is identifying relevant portions of the input, while a low CE implies that the model classifies these components as motifs correctly.

## E  PROOF OF MOTIF IDENTIFIABILITY THEOREM

The following is a formal proof of the Motif Identifiability Theorem. The term $\#^*_{\max}$ is used in this proof to denote $\max_c \#^*_c$

### E.1  PROOF SKETCH

We proceed by contrapositive, starting with the assumption that $\mathcal{E}_m(\hat{g}) \geq k\epsilon$ and then proving that $\mathcal{E}(\hat{h} \circ \hat{g}) \geq \epsilon$. We first demonstrate (Lemma E.2.1) that high motif error implies either a high number of false negatives for some channel $c$ (true motif cells that have no coverage by $\hat{g}$) or simultaneously a low number of false positives and some channels $c_1, c_2$ such that there is some high number of cells of both that are covered by the same channel $c'$ of $\hat{g}$. This theorem is proven by a simple counting argument, relying only on the fact that $\delta(\hat{g}) = \delta^*$. We then prove in each of the two resulting cases that the property holds for some fraction of motif cells in general, using MOTIF-SUFFICIENCY and NON-OVERLAPPING. We then apply $\alpha$-MOTIF-NECESSITY to each case, demonstrating that $\hat{g}$ does not distinguish different inputs that must lead to different values of $y^* = h^*(g^*(x))$. Since $\hat{g}$ cannot distinguish these inputs, neither can $\hat{h} \circ \hat{g}$, and thus in one of the two cases error must arise. Thus, in both cases, we conclude that $\mathcal{E}(\hat{h} \circ \hat{g}) \geq \epsilon$.

### E.2  LEMMAS

#### E.2.1  SOURCES OF MOTIF ERROR DICHOTOMY

First, we define a few quantities, representing the number of times a true cell is covered negatively, covered once, or covered multiple times.

$$\text{FN}_g(c) = \mathbb{E}_{x,m^*} \left[ \sum_{i \in \omega_c(m^*)} \mathbf{1}(v_{\hat{g}(x)}(i) = 0) \right]$$

$$\text{CO}_g(c, c') = \mathbb{E}_{x,m^*} \left[ \sum_{i \in \omega_c(m^*)} \mathbf{1}\left(v_{\hat{g}(x)}(i) = 1 \wedge v_{\hat{g}(x)}(i, c') = 1\right) \right]$$

$$\text{CM}_g(c) = \mathbb{E}_{x,m^*} \left[ \sum_{i \in \omega_c(m^*)} \mathbf{1}\left(v_{\hat{g}(x)}(i) > 1\right) \right]$$

We also define the quantity

$$\text{FP}_g = \mathbb{E}_{x,m^*} \left[\text{FP}_g(x)\right]$$

where

$$\text{FP}_g(x) = \sum_{c'} \sum_{i \in I \setminus \bigcap_c \omega_c(m^*)} \mathbf{1}(\hat{g}(x)[i, c'] = 1)$$

The claim we wish to establish is

$$\forall \hat{h} \in M \rightarrow Y, \hat{g} \in \mathcal{G}, \delta(\hat{g}) = \delta^* \wedge \mathcal{E}_m(\hat{g}) \geq k\epsilon$$
$$\implies (\exists c, \text{FN}_g(c) \geq \beta_1)$$
$$\vee (\exists c_1, c_2, c', \min(\text{CO}_g(c_1, c'), \text{CO}_g(c_2, c')) \geq \beta_2) \wedge (FP_g \leq n\beta_1)$$

For

$$\beta_2 = \frac{\#^* k \epsilon}{2n(n-1)}$$

and

$$\beta_1 = \frac{\alpha \beta_2}{4 \#_{\max} |\Delta|}$$

We proceed by contrapositive, assuming that $(\forall c, \text{FN}_g(c) < \beta_1)$ and $(\forall c_1, c_2, c', \min(\text{CO}_g(c_1, c'), \text{CO}_g(c_2, c')) < \beta_2) \vee (FP_g > n\beta_1)$ both hold. Note that this proof relies on none of our assumptions and is just about counting the outputs of $\hat{g}$.

**Bounding CM and FP** First, we bound CM and FP. Specifically, we establish that

$$\sum_{c,c'} \text{CO}_g(c,c') = \sum_c \mathbb{E}_{x,m^*}\left[ \sum_{i \in \omega_c(m^*)} \sum_{c'} \mathbf{1}\left(v_{\hat{g}(x)}(i) = 1 \wedge v_{\hat{g}(x)}(i, c') = 1\right)\right]$$

$$= \sum_c \mathbb{E}_{x,m^*}\left[ \sum_{i \in \omega_c(m^*)} \mathbf{1}(v_{\hat{g}(x)}(i) = 1)\right]$$

$$\sum_{c,c'} \text{CO}_g(c,c') + 2\sum_c \text{CM}_g(c) = \sum_c \mathbb{E}_{x,m^*}\left[ \sum_{i \in \omega_c(m^*)} \mathbf{1}(v_{\hat{g}(x)}(i) = 1) + 2 \cdot \mathbf{1}\left(v_{\hat{g}(x)}(i) > 1\right)\right]$$

$$\leq \sum_c \mathbb{E}_{x,m^*}\left[ \sum_{i \in \omega_c(m^*)} v_{\hat{g}(x)}(i)\right]$$

$$= \sum_c \mathbb{E}_{x,m^*}\left[ \sum_{i \in I} \hat{g}(x)[i, c]\right] - \text{FP}_g$$

$$= \#(\hat{g}) - \text{FP}_g$$

$$\text{FP}_g + \sum_{c,c'} \text{CO}_g(c,c') + 2\sum_c \text{CM}_g(c) \leq \#*$$

$$\sum_c \text{FN}_g(c) + \sum_{c,c'} \text{CO}_g(c,c') + \sum_c \text{CM}_g(c) = \sum_c \mathbb{E}_{x,m^*}\left[ \sum_{i \in \omega_c(m^*)} 1\right]$$

$$= \#^*$$

$$\sum_c \text{FN}_g(c) + \sum_{c,c'} \text{CO}_g(c,c') + \sum_c \text{CM}_g(c) \geq \text{FP}_g + \sum_{c,c'} \text{CO}_g(c,c') + 2\sum_c \text{CM}_g(c)$$

$$\sum_c \text{FN}_g(c) \geq \text{FP}_g + \sum_c \text{CM}_g(c)$$

And thus we have a bound on CM and FP in terms of FN. Note that this means we can eliminate the $\text{FP}_g > n\beta_1$ disjunction from our premises as we now know that $\text{FP}_g \leq \sum_c \text{FN}_g(c) \leq n\beta_1$.

**Low FN implies high CO** From above we have

$$\sum_c \text{FN}_g(c) + \sum_{c,c'} \text{CO}_g(c,c') + \sum_c \text{CM}_g(c) = \#^*$$

From this and the previous result it is clear that

$$2\sum_c \text{FN}_g(c) + \sum_{c,c'} \text{CO}_g(c,c') \geq \#^*$$

We then can state

$$\sum_{c,c'} \text{CO}_g(c,c') \geq \#^* - 2\sum_c \text{FN}_g(c)$$

**High CO implies low $\mathcal{E}_m$**  We now define the following function $\pi : [n] \to [n]$ assigning the "proper channel" of a given channel of $\hat{g}$ as

$$\pi(c') = \arg\max_c CO_g(c, c')$$

Assume that $\forall c_1, c_2, c', \min(CO_g(c_1, c'), CO_g(c_2, c')) \leq \beta_2$. We then have that

$$\forall c \neq \pi(c'), CO_g(c, c') \leq \min(CO_g(c, c'), CO_g(\pi(c'), c')) \leq \beta_2$$

Finally, we have that

$$\sum_{c,c'} CO_g(c, c') \leq n(n-1)\beta_2 + \sum_{c'} CO_g(\pi(c'), c')$$

We then express

$$\sum_{c'} CO_g(\pi(c'), c') \leq \sum_c \sum_{c'|\pi(c')=c} CO_g(c, c')$$

$$= \sum_c \sum_{c'|\pi(c')=c} \mathbb{E}_{x,m^*}\left[\sum_{i\in\omega_c(m^*)} \mathbf{1}\left(v_{\hat{g}(x)}(i) = 1 \wedge v_{\hat{g}(x)}(i, c') = 1\right)\right]$$

$$\leq \#^* - \#^*\mathcal{E}_m(\hat{g})$$

where the last step is viable as $\max(\#^*, \#(\hat{g})) = \#^*$ as $\delta(\hat{g}) = \delta^*$ We thus have that

$$\sum_{c,c'} CO_g(c, c') \leq n(n-1)\beta_2 + \#^* - \#^*\mathcal{E}_m(\hat{g})$$

and therefore

$$\#^*\mathcal{E}_m(\hat{g}) \leq n(n-1)\beta_2 + \#^* - \sum_{c,c'} CO_g(c, c')$$

**Final proof**  We can then add the assumption $\forall c, \text{FN}(c) \leq \beta_1$. This means that

$$\sum_{c,c'} CO_g(c, c') \geq \#^* - 2n\beta_1$$

Putting this together with the above, we have

$$\#^*\mathcal{E}_m(\hat{g}) \leq n(n-1)\beta_2 + 2n\beta_1 \leq n(n-1)\beta_2 + n\beta_2 = (n(n-1)+n)\beta_2 \leq 2n(n-1)\beta_2 = \#^*k\epsilon$$

Thus ending our proof

### E.2.2  COROLLARY: MOTIF ERROR AT ALL POSITIONS

We define the extended footprint of a cell as a function $\phi : I \to 2^{I \times [d]}$ mapping a location to the set of locations in the input whose output is in $p_2(i)$

$$\phi(i) = \{i'' : \exists i' \in p_2(i), i'' \in p(i')\}$$

Now, we establish that motif error in some percentage of positions implies a consistent probability of motif error every time the motif shows up, regardless of skeleton. First, define $\tilde{P}_{c,i}$ to be a distribution over regions of size $\phi(i)$ defined as

$$\tilde{P}_{c,i}(\eta) = P[x[\phi(i)] = \eta | g^*(x)[i, c] \neq 0]$$

We can use MOTIF-SUFFICIENCY to break this down as (letting $o$ be the relative position of $i$ within $\eta$

$$\tilde{P}_{c,i}(\eta) = P[x[\phi(i)] = \eta | g^*(x)[i, c] \neq 0]$$
$$= P_c[\eta[p(i) - o]]P_b[x[\phi(i) \setminus p(i)] = \eta[(\phi(i) \setminus p(i)) - o]]$$

We implicitly use NON-OVERLAPPING when we assume that $\phi(i) \setminus p(i)$ is entirely over the background. The specific property here is that $\phi(i) \cap p(i') = \emptyset$ for all $i, i' \in m^*$, $i' \neq i$. This follows from NON-OVERLAPPING as we have that

$$
\begin{aligned}
\phi(i) \cap p(i') \neq \emptyset &\iff \exists i'', i'' \in \phi(i) \cap p(i') \\
&\iff \exists i'', i'' \in \phi(i) \wedge i'' \in p(i') \\
&\iff \exists i'', (\exists j \in p_2(i), i'' \in p(j)) \wedge i'' \in p(i') \\
&\iff \exists j, j \in p_2(i) \wedge \exists i'', i'' \in p(j) \wedge i'' \in p(i') \\
&\iff \exists j, j \in p_2(i) \wedge (p(j) \cap p(i')) \neq \emptyset \\
&\iff \exists j, j \in p_2(i) \wedge j \in p_2(i') \\
&\iff p_2(i) \cap p_2(i') \neq \emptyset
\end{aligned}
$$

Which is the exact condition given in NON-OVERLAPPING.

Note that this is no longer in any way dependent on $i$ due to the translational invariance of $P_b$. Therefore, we have that $\tilde{P}_{c,i}(\eta) = \tilde{P}_c(\eta)$, and this is consistent at all locations that $c$ appears, regardless of the skeleton.

We also define $q : 2^n \times \Delta \to 2^{\Delta \times [n]}$ be be a function that takes a vector $u$ and offset $d$ and returns the map $q(u, d)$ such that $q(u, d)[d] = u \wedge \forall d' \neq d, q(u, d)[d'] = 0$. Let $Q(u) = \{q(u, d) : d \in \Delta\}$

**Claim** The claim we wish to establish is

$$
\forall \hat{h} \in M \to Y, \hat{g} \in \mathcal{G}, \delta(\hat{g}) = \delta^* \wedge \mathcal{E}_m(\hat{g}) \geq k\epsilon
$$
$$
\implies \left( \exists c, P\left[\hat{g}(\eta) = 0 | \eta \sim \tilde{P}_c\right] \geq \frac{\beta_1}{\#_{\max}} \right)
$$
$$
\vee \Big(
$$
$$
\left( \exists c_1, c_2, c', \min_{c \in \{c_1, c_2\}} P\left[\hat{g}(\eta) \in Q(\mathbf{e}_{c'}) | \eta \sim \tilde{P}_c\right] \geq \frac{\beta_2}{\#_{\max}} \right)
$$
$$
\wedge
$$
$$
(FP_g \leq n\beta_1)
$$
$$
\Big)
$$

**False negative case** We now start with the assumption that $\text{FN}_{\hat{g}}(c) \geq \beta$. We have that

$$
\begin{aligned}
\text{FN}_g(c) &= \mathbb{E}_{x,m^*}\left[ \sum_{i \in \omega_c(m^*)} \mathbf{1}(v_{\hat{g}(x)}(i) = 0) \right] \\
&= \sum_m \mathbb{E}_{x,m^*}\left[ \sum_{i \in \omega_c(m^*)} \mathbf{1}(v_{\hat{g}(x)}(i) = 0) | g^*(x) = m \right] P_m(m) \\
&= \sum_m \sum_{i \in m} \mathbb{E}_{x,m^*}\left[ \mathbf{1}(v_{\hat{g}(x)}(i) = 0) | g^*(x) = m \right] P_m(m) \\
&= \sum_m \sum_{i \in m} P\left[ \mathbf{1}(v_{\hat{g}(x)}(i) = 0) | g^*(x) = m \right] P_m(m) \\
&= \sum_m \sum_{i \in m} P\left[ \hat{g}(\eta) = 0 | \eta \sim \tilde{P}_c \right] P_m(m) \\
&= P\left[ \hat{g}(\eta) = 0 | \eta \sim \tilde{P}_c \right] \sum_m \sum_{i \in m} P_m(m) \\
&= P\left[ \hat{g}(\eta) = 0 | \eta \sim \tilde{P}_c \right] \mathbb{E}\left[ \sum_{i \in m} 1 \right] \\
&= P\left[ \hat{g}(\eta) = 0 | \eta \sim \tilde{P}_c \right] \#_c
\end{aligned}
$$

and thus we can conclude that $P\left[\hat{g}(\eta) = 0 | \eta \sim \tilde{P}_c\right] \geq \frac{\beta_1}{\#_c} \geq \frac{\beta_1}{\#_{\max}}$.

**Confusion Case**

In this case, we have two properties, first that we have some $c_1$ and $c_2$ such that

$$\mathrm{CO}_g(c_1, c') \geq \beta_2 \wedge \mathrm{CO}_g(c_2, c') \geq \beta_2$$

and the second that

$$\mathrm{FP}_g \leq \beta_1$$

First, we use a similar argument to the previous case to establish that

$$\mathrm{CO}_g(c, c') = \mathbb{E}_{x,m^*}\left[\sum_{i \in \omega_c(m^*)} \mathbf{1}(v_{\hat{g}(x)}(i) = 1 \wedge v_{\hat{g}(x)}(i, c') = 1)\right]$$

$$= \sum_m \mathbb{E}_{x,m^*}\left[\sum_{i \in \omega_c(m^*)} \mathbf{1}(v_{\hat{g}(x)}(i) = 1 \wedge v_{\hat{g}(x)}(i, c') = 1 | g^*(x) = m\right] P_m(m)$$

$$= \sum_m \sum_{i \in m} \mathbb{E}_{x,m^*}\left[\mathbf{1}(v_{\hat{g}(x)}(i) = 1 \wedge v_{\hat{g}(x)}(i, c') = 1 | g^*(x) = m\right] P_m(m)$$

$$= \sum_m \sum_{i \in m} P\left[\mathbf{1}(v_{\hat{g}(x)}(i) = 1 \wedge v_{\hat{g}(x)}(i, c') = 1) | g^*(x) = m\right] P_m(m)$$

$$= \sum_m \sum_{i \in m} P\left[\hat{g}(\eta) \in Q(\mathbf{e}_{c'}) | \eta \sim \tilde{P}_c\right] P_m(m)$$

$$= P\left[\hat{g}(\eta) \in Q(\mathbf{e}_{c'}) | \eta \sim \tilde{P}_c\right] \sum_m \sum_{i \in m} P_m(m)$$

$$= P\left[\hat{g}(\eta) \in Q(\mathbf{e}_{c'}) | \eta \sim \tilde{P}_c\right] \mathbb{E}\left[\sum_{i \in m} 1\right]$$

$$= P\left[\hat{g}(\eta) \in Q(\mathbf{e}_{c'}) | \eta \sim \tilde{P}_c\right] \#_c$$

and thus we can conclude that $P\left[\hat{g}(\eta) \in Q(\mathbf{e}_{c'}) | \eta \sim \tilde{P}_c\right] \geq \frac{\beta_2}{\#_c} \geq \frac{\beta_2}{\#_{\max}}$ for $c \in \{c_1, c_2\}$.

### E.2.3 LEMMA: INDISTINGUISHABLE LOCAL-TO-GLOBAL

Statement: given a pairing scheme $\psi$, a predicate $\zeta : \mathbb{R}^{I \times [d]} \to \mathbb{B}$, some $\kappa > 0$, and that for all $\psi(m_2 | m_1) > 0$ that are an $i$-OFF-BY-ONE PAIR and for all $x_R \in \mathbb{R}^{I \times [d] \setminus \phi(i)}$ we can assume

$$P[\zeta(x) | m_1, x_R] + P[\zeta(x) | m_2, x_R] \geq \kappa$$

we can prove that

$$P[\zeta(x)] \geq \frac{1}{2}\alpha\kappa$$

We begin by multiplying by $P[x_R | m_1] = P[x_R | m_2]$ (these are equal because $m_1$ and $m_2$ agree outside of $\phi(i)$)

$$P[\zeta(x) | m_1, x_R] P[x_R | m_1] + P[\zeta(x) | m_2, x_R] P[x_R | m_2] \geq \kappa P[x_R | m_1]$$
$$P[\zeta(x), x_R | m_1] + P[\zeta(x), x_R | m_2] \geq \kappa P[x_R | m_1]$$

and integrating

$$\int P[\zeta(x), x_R | m_1] \mathrm{d}x_R + \int P[\zeta(x), x_R | m_2] \mathrm{d}x_R \geq \kappa \int P[x_R | m_1] \mathrm{d}x_R$$
$$P[\zeta(x) | m_1] + P[\zeta(x) | m_2] \geq \kappa$$

We then multiply both sides by $\psi(m_2|m_1)P_m(m_1)$ and sum:

$$\sum_{m_1,m_2\in M} \psi(m_2|m_1)P_m(m_1)(P[\zeta(x)|m_1] + P[\zeta(x)|m_2]) \geq \sum_{m_1,m_2\in M} \psi(m_2|m_1)P_m(m_1)\kappa$$

We have that

$$\begin{aligned}
\text{LHS} &= \sum_{m_1,m_2\in M} \psi(m_2|m_1)P_m(m_1)(P[\zeta(x)|m_1] + P[\zeta(x)|m_2]) \\
&= \sum_{m_1,m_2\in M} \psi(m_2|m_1)P_m(m_1)P[\zeta(x)|m_1] + \sum_{m_1,m_2\in M} \psi(m_2|m_1)P_m(m_1)P[\zeta(x)|m_2] \\
&= \sum_{m_2\in M} \psi(m_2|m_1)P[\zeta(x)] + \sum_{m_2\in M} q(m_2)P[\zeta(x)|m_2] \\
&\leq P[\zeta(x)] + \sum_{m_2\in M} P_m(m_2)P[\zeta(x)|m_2] \\
&\leq P[\zeta(x)] + P[\zeta(x)]
\end{aligned}$$

$$P[\zeta(x)] \geq \frac{1}{2}\text{LHS}$$

$$\begin{aligned}
\text{RHS} &= \sum_{m_1,m_2\in M} \psi(m_2|m_1)P_m(m_1)\kappa \\
&= \sum_{m_2\in M} q(m_2)\kappa \\
&\geq \alpha\kappa
\end{aligned}$$

and therefore, we have that

$$P[\zeta(x)] \geq \frac{1}{2}\kappa\alpha$$

which completes our proof.

### E.2.4 Lemma: False negatives

Given some $\hat{g}$, $\kappa < \frac{1}{2}$ such that

$$P[\hat{g}(\eta_1) = 0|\eta_1 \sim \tilde{P}_c] \geq \kappa$$

we have that for all $\hat{h}$,

$$\mathcal{E}(\hat{h} \circ \hat{g}) \geq \frac{1}{2}\alpha\kappa$$

We now proceed with our proof. Let $\psi$ be the pairing scheme corresponding to $v_1 = \mathbf{e}_c$ and $v_2 = \mathbf{0}$, and $d_1 = d_2 = 0$. Fix any $m_1, m_2, i$ such that $\psi(m_2|m_1) > 0$ being an $i$-OFF-BY-ONE PAIR and $x_R \in \mathbb{R}^{I\times[d]\setminus\phi(i)}$. We can now see that

$$\begin{aligned}
P[\hat{f}(x) \neq y^*|m_1, x_R] &\geq P[\hat{f}(x) \neq y^*, \hat{g}(\phi(i)) = 0|m_1, x_R] \\
&= P[\hat{f}(x) \neq y^*|\hat{g}(\phi(i)) = 0, m_1, x_R]P[\hat{g}(\phi(i)) = 0|m_1, x_R] \\
&= P[\hat{f}(x) \neq y^*|\hat{g}(\phi(i)) = 0, m_1, x_R]P[\hat{g}(\eta_1) = 0|\eta_1 \sim \tilde{P}_c] \\
&\geq P[\hat{f}(x) \neq y^*|\hat{g}(\phi(i)) = 0, m_1, x_R]\kappa
\end{aligned}$$

Once we know $x_R$ and that $\hat{g}(\phi(i)) = 0$, we know that $\hat{f}(x)$ is entirely dependent on $x_R$ and not on $x[p(i)]$ because we have access via $\hat{g}(\phi(i))$ to all values of $\hat{g}(x)$ that are influenced by $x[p(i)]$. As such, we can replace $\hat{f}(x)$ with $\lambda(x_R)$. Thus, we have

$$P[\hat{f}(x) \neq y^*|m_1, x_R] \geq P[\lambda(x_R) \neq y^*|m_1, x_R]\kappa$$

By the translational property of $P_b$ we know that $P[\hat{g}(\eta_1) = 0|\eta_1 \sim P_b]$ is a definable, fixed, quantity, and applies to any random variable $\eta_i = x[\phi(i)]$. We thus have that $P[\hat{g}(\eta_1) = 0|\eta_1 \sim$

$P_b] \geq 1 - n\delta$ as otherwise $\hat{g}$ would not be capable of having a density of $\delta$ on the whole image. We can safely assume $n\delta < \frac{1}{2}$ since otherwise the NON-OVERLAPPING would be violated, since the minimum size of a motif cell is 3 (1-dimensional, radius 1). Thus we can assume that

$$P[\hat{g}(\eta_1) = 0 | \eta_1 \sim P_b] \geq \frac{1}{2} \geq \kappa$$

and thus have the same property

$$P[\hat{f}(x) \neq y^* | m_2, x_R] \geq P[\lambda(x_R) \neq y^* | m_2, x_R]\kappa$$

We have that $h^*(m_1) \neq h^*(m_2)$. We can then proceed

$$
\begin{aligned}
P[\hat{f}(x) \neq y^* | m_1, x_R] + P[\hat{f}(x) \neq y^* | m_2, x_R] &\geq \kappa(P[\lambda(x_R) \neq y^* | m_1, x_R] + P[\lambda(x_R) \neq y^* | m_2, x_R]) \\
&\geq \kappa(P[\lambda(x_R) \neq h^*(m_1) | m_1, x_R] + P[\lambda(x_R) \neq h^*(m_2) | m_2, x_R]) \\
&= \kappa(\mathbf{1}(\lambda(x_R) \neq h^*(m_1)) + \mathbf{1}(\lambda(x_R) \neq h^*(m_2)) \\
&\geq \kappa(\mathbf{1}(\lambda(x_R) \neq h^*(m_1) \vee \lambda(x_R) \neq h^*(m_2)) \\
&= \kappa
\end{aligned}
$$

As such, we can now apply Lemma E.2.3 to get the statement

$$P[\hat{f}(x) \neq y^*] \geq \frac{1}{2}\kappa\alpha$$

which completes our proof

### E.2.5 LEMMA: CONFUSION

Given $c_1$, $c_2$, and $c'$, $\kappa$, and some $\hat{g}$ such that

$$P[\hat{g}(\eta_1) \in Q(\mathbf{e}_{c'}) | \eta_1 \sim \tilde{P}_{c_1}] \geq \kappa \wedge P[\hat{g}(\eta_2) \in Q(\mathbf{e}_{c'}) | \eta_2 \sim \tilde{P}_{c_2}] \geq \kappa$$

we have that for all $\hat{h}$

$$\mathbb{E}[\hat{f}(x) \neq y^* \vee \mathrm{FP}_g(x) > 0] \geq \frac{\alpha\kappa}{2|\Delta|}$$

We now proceed with our proof.

We proceed as in the proof of Lemma E.2.4, with a few variations. Consider the pairing scheme $\psi$ corresponding to $v_1 = \mathbf{e}_{c_1}$ and $v_2 = \mathbf{e}_{c_2}$, and $d_1 = -\arg\max_d P[\hat{g}(\eta_1) = q(u, d) | \eta_1 \sim \tilde{P}_{v_1}]$ and $d_2 = -\arg\max_d P[\hat{g}(\eta_2) = q(u, d) | \eta_2 \sim \tilde{P}_{v_2}]$. We have that $P[\hat{g}(\eta_1) = q(u, -d_1) | \eta_1 \sim \tilde{P}_{v_1}], P[\hat{g}(\eta_2) = q(u, -d_2) | \eta_2 \sim \tilde{P}_{v_2}] \geq \kappa/|\Delta|$. Let $\kappa' = \kappa/|\Delta|$

Let $(m_1, m_2)$ be an $i$-OFF-BY-ONE PAIR such that $\psi(m_2 | m_1) > 0$. Fix $x_R \in \mathbb{R}^{I \times [d] \setminus \phi(i)}$. Consider

$$P[\hat{f}(x) \neq y^* \vee \mathrm{FP}_g(x) > 0 | m_1, x_R] + P[\hat{f}(x) \neq y^* \vee \mathrm{FP}_g(x) > 0 | m_2, x_R]$$

We have that $h^*(m_1) \neq h^*(m_2)$. We also know that

$$P[\hat{f}(x) \neq y^* \vee \mathrm{FP}_g(x) > 0 | m_1, x_R] \geq P[\hat{f}(x) \neq h^*(m_1) \vee \mathrm{FP}_g(x) > 0 | m_1, x_R]$$

We can then analyze

$$
\begin{aligned}
P[\hat{f}(x) \neq h^*(m_1) \vee \mathrm{FP}_g(x) > 0 | m_1, x_R] &\geq P[\hat{f}(x) \neq h^*(m_1) \vee \mathrm{FP}_g(x) > 0, \hat{g}(x[\phi(i - d_1)]) = q(u, 0) | m_1, x_R] \\
&= P[\hat{f}(x) \neq h^*(m_1) \vee \mathrm{FP}_g(x) > 0 | m_1, x_R, x[\phi(i - d_1)] = q(u, 0)]P[x[\phi(i - d_1)] \\
&\geq P[\hat{f}(x) \neq h^*(m_1) \vee \mathrm{FP}_g(x) > 0 | m_1, x_R, \hat{g}(x[\phi(i - d_1)]) = q(u, 0)]\kappa'
\end{aligned}
$$

where the last step comes from the fact that $m_1$ has its motif at $i + d_1$, and therefore, $\hat{g}$ should activate at $i$. Finally, if we define $\lambda(x_R)$ to be the value $\hat{h}(\hat{m})$ takes when $\hat{m}[i'] = \hat{g}(x[\phi(i')])$ for all $i'$ in a motif cell of $m^*$ other than $i$ and 0 otherwise, we have that

$$\hat{f}(x) \neq \lambda(x_R) \implies \mathrm{FP}_g(x) > 0$$

because if it is equal to any other value, that indicates that $\hat{g}$ is sending some values through non-motif cell channels. We thus have that

$$\hat{f}(x) \neq h^*(m_1) \vee \mathrm{FP}_g(x) > 0 \iff \lambda(x_R) \neq h^*(m_1) \vee \mathrm{FP}_g(x) > 0$$

Thus, we have that

$$P[\hat{f}(x) \neq h^*(m_1) \vee \text{FP}_g(x) > 0|m_1, x_R] \geq \kappa' P[\lambda(x_R) \neq h^*(m_1) \vee \text{FP}_g(x) > 0|m_1, x_R]$$

and by an identical argument

$$P[\hat{f}(x) \neq h^*(m_2) \vee \text{FP}_g(x) > 0|m_2, x_R] \geq \kappa' P[\lambda(x_R) \neq h^*(m_2) \vee \text{FP}_g(x) > 0|m_2, x_R]$$

We now proceed by cases. Either $\lambda(x_R) \neq h^*(m_1)$, in which case

$$P[\lambda(x_R) \neq h^*(m_1) \vee \text{FP}_g(x) > 0|m_1, x_R] = 1$$

or $\hat{f}(x) = h^*(m_1)$ and thus $\lambda(x_R) \neq h^*(m_2)$ and thus

$$P[\lambda(x_R) \neq h^*(m_2) \vee \text{FP}_g(x) > 0|m_2, x_R] = 1$$

In either case, we have

$$P[\hat{f}(x) \neq y^* \vee \text{FP}_g(x) > 0|m_1, x_R] + P[\hat{f}(x) \neq y^* \vee \text{FP}_g(x) > 0|m_2, x_R] \geq \kappa'$$

Applying Lemma E.2.3 to this statement, we get that we have

$$P[\hat{f}(x) \neq y^* \vee \text{FP}_g(x) > 0] \geq \frac{1}{2}\kappa'\alpha$$

completing our proof

### E.3    MAIN PROOF

The statement is reproduced below:

$$\forall \hat{g} \in \mathcal{G}, \hat{h} \, . \, \delta(\hat{g}) = \delta^* \implies \left( \forall \epsilon > 0, \mathcal{E}(\hat{h} \circ \hat{g}) < \epsilon \implies \mathcal{E}_m(\hat{g}) < k\epsilon \right)$$

Let

$$k = \frac{16\#_{\max}^2 |\Delta| n(n-1)}{\#^* \alpha^2}$$

and then fix $\hat{g}$ such that $\delta(\hat{g}) = \delta^*$ and $\epsilon > 0$. Assume towards contradiction that the statement $\mathcal{E}(\hat{h} \circ \hat{g}) < \epsilon \implies \mathcal{E}_m(\hat{g}) < k\epsilon$ is false. We then have $\mathcal{E}(\hat{h} \circ \hat{g}) < \epsilon$ and $\mathcal{E}_m(\hat{g}) \geq k\epsilon$. Using Corrolary E.2.2 we have two cases.

#### E.3.1    FALSE NEGATIVE CASE

We have that there is some $c$ for which

$$P\left[\hat{g}(\eta) = 0|\eta \sim \tilde{P}_c\right] \geq \frac{\beta_1}{\#_{\max}}$$

Applying Lemma E.2.4, we have that

$$\mathcal{E}(\hat{h} \circ \hat{g}) \geq \frac{1}{2}\alpha\frac{\beta_1}{\#_{\max}} = \frac{\alpha^2\beta_2}{8\#_{\max}^2|\Delta|} = \frac{\alpha^2\#^* k\epsilon}{16\#_{\max}^2|\Delta|n(n-1)} = \epsilon$$

which is a contradiction with $\mathcal{E}(\hat{h} \circ \hat{g}) < \epsilon$.

#### E.3.2    CONFUSION CASE

We have that there exist some $c_1, c_2, c'$ such that

$$\min_{c \in \{c_1, c_2\}} P\left[\hat{g}(\eta) \in Q(\mathbf{e}_{c'})|\eta \sim \tilde{P}_c\right] \geq \frac{\beta_2}{\#_{\max}}$$

and also,

$$FP_g \leq n\beta_1$$

Applying Lemma E.2.5, we have that

$$\mathbb{E}[\hat{f}(x) \neq y^* \vee \text{FP}_g(x)] \geq \frac{1}{2}\alpha\frac{\beta_2}{|\Delta|\#_{\max}}$$

We also know that

$$\mathbb{E}[\text{FP}_g(x)] \leq \beta_1$$

and therefore

$$\mathcal{E}(\hat{f}) \geq \frac{1}{2}\alpha\frac{\beta_2}{|\Delta|\#_{\max}} - \beta_1 = \frac{1}{4}\alpha\frac{\beta_2}{|\Delta|\#_{\max}} = \frac{1}{8}\alpha\frac{\#^* k\epsilon}{n(n-1)|\Delta|\#_{\max}} = \frac{2\#_{\max}\epsilon}{\alpha} > \epsilon$$

which is a contradiction with $\mathcal{E}(\hat{h} \circ \hat{g}) < \epsilon$, thus completing our proof.

# F  MOTIF ERROR EQUIVALENCE

In this section, we prove that our proof's error metric is only ever off by a constant factor from our empirical error metrics.

## F.1  FORMAL STATEMENT

For all $\hat{g} \in \mathcal{G}$ such that $\delta(\hat{g}) = \delta^*$,

$$\mathcal{E}_m(\hat{g}) \le \epsilon \implies \text{FNE}(\hat{g}) \le \epsilon \wedge \text{FPE}(\hat{g}) \le \epsilon \wedge \text{CE}(\hat{g}) \le 2\epsilon$$

$$\mathcal{E}_m(\hat{g}) \le \epsilon \impliedby \text{FNE}(\hat{g}) \le \frac{1}{4}\epsilon \wedge \text{CE}(\hat{g}) \le \frac{1}{2}\epsilon$$

## F.2  CORRESPONDENCE WITH QUANTITIES FROM LEMMA E.2.1

First, note that

$$\#^* \mathcal{E}_m(\hat{g}) = \min_\tau \#^* - \sum_c \sum_{c' | c = \tau(c')} \text{CO}_g(c, c')$$

Then, note that

$$\text{FNE}(\hat{g}) = \frac{\sum_c \text{FN}_g(c)}{\#^*}$$

$$\text{FPE}(\hat{g}) = \frac{\text{FP}_g}{\#(\hat{g})}$$

by inspection. The case of CE is more complicated, due to the presence of MM. Inspecting the denominator, we have

$$|\text{MM}(\hat{m}, m^*)| + |\text{NMM}(\hat{m}, m^*)| = |P(\hat{m})| - |\text{FPM}(\hat{m}, m^*)|$$

and therefore

$$\mathbb{E}[|\text{MM}(\hat{g}(x), g^*(x))|] + \mathbb{E}[|\text{NMM}(\hat{g}(x), g^*(x))|] = \#(\hat{g})(1 - \text{FPE}(\hat{g}))$$

Additionally, we can note that if we assume there are no ties in the $\max$ computation (or alternatively, they are broken in some systematic way rather than leading to duplicates), we know that

$$|\text{MM}(\hat{m}, m^*)| = |\{(i, c) \in P(g^*(x)) : \exists (i', c') \in P(\hat{g}(x)) : i' \in p_2(i)\}|$$

and thus

$$\mathbb{E}[|\text{MM}(\hat{g}(x), g^*(x))|] = \#^*(1 - \text{FNE}(\hat{g}))$$

Letting $Q(\hat{m}, m^*) = \{(i, c) \in P(g^*(x)) : \exists (i', c') \in P(\hat{g}(x)) : i' \in p_2(i)\}$ we can break this down into a dichotomy

$$Q(\hat{m}, m^*) = Q_1(\hat{m}, m^*) \sqcup Q_2(\hat{m}, m^*)$$

where

$$Q_1(\hat{m}, m^*) = \{(i, c) \in P(g^*(x)) : \exists!(i', c') \in P(\hat{g}(x)) : i' \in p_2(i)\}$$
$$Q_2(\hat{m}, m^*) = \{(i, c) \in P(g^*(x)) : \exists (i'_1, c'_1) \ne (i'_2, c'_2) \in P(\hat{g}(x)) : i'_1, i'_2 \in p_2(i)\}$$

We have that

$$\mathbb{E}[|Q_2(\hat{g}(x), g^*(x))|] = \sum_c \text{CM}_g(c)$$

Finally, we can see that

$$|\mathrm{conf}_\tau(\hat{m}, m^*)| = \sum_{(i,c)\in Q} |\{(i', c') \in \mathrm{conf}_\tau(\hat{m}, m^*) : i' \in p_2(i)\}|$$

$$= \lambda_\tau(\hat{m}, m^*)|Q_2(\hat{m}, m^*)| + \sum_{(i,c)\in Q_1} |\{(i', c') \in \mathrm{conf}_\tau(\hat{m}, m^*) : i' \in p_2(i)\}|$$

$$= \lambda_\tau(\hat{m}, m^*)|Q_2(\hat{m}, m^*)| + \sum_{(i,c)\in Q_1} \mathbf{1}(\exists c', \tau(c') = c \wedge v_m(i) = 1 \wedge v_m(i, c') \neq 1)$$

$$= \lambda_\tau(\hat{m}, m^*)|Q_2(\hat{m}, m^*)| + \sum_{(i,c)\in Q_1} \sum_{c'|\tau(c')=c} \mathbf{1}(v_m(i) = 1 \wedge v_m(i, c') = 1)$$

$$\mathbb{E}[|\mathrm{conf}_\tau(\hat{g}(x), g^*(x))|] = \lambda_\tau \sum_c \mathrm{CM}_g(c) + \sum_c \sum_{c'|\tau(c')=c} \mathrm{CO}_g(c, c')$$

$$= \lambda_\tau \sum_c \mathrm{CM}_g(c) + \sum_{c,c'} \mathrm{CO}_g(c, c') - \sum_c \sum_{c'|\tau(c')=c} \mathrm{CO}_g(c, c')$$

$$= \lambda_\tau \sum_c \mathrm{CM}_g(c) + \#^* - \sum_c \mathrm{FN}_g(c) - \sum_c \mathrm{CM}_g(c) - \sum_c \sum_{c'|\tau(c')=c} \mathrm{CO}_g(c, c')$$

where $\lambda_\tau(\hat{m}, m^*)$ and $\lambda_\tau$ are some constants in $[0, 1]$.

Since $(\min_x A(x)) + (\min_x B(x)) \leq \min_x(A(x) + B(x)) \leq (\min_x A(x)) + (\max_x B(x))$, we have that

$$\min_\tau \mathbb{E}[|\mathrm{conf}_\tau(\hat{g}(x), g^*(x))|] = \#^* \mathcal{E}_m(\hat{g}) + \lambda_1 \sum_c \mathrm{CM}_g(c) - \sum_c \mathrm{FN}_g(c) - \sum_c \mathrm{CM}_g(c)$$

for some $\lambda_1 \in [0, 1]$ and thus

$$\min_\tau \mathbb{E}[|\mathrm{conf}_\tau(\hat{g}(x), g^*(x))|] = \#^* \mathcal{E}_m(\hat{g}) - \lambda_2 \sum_c \mathrm{CM}_g(c) - \sum_c \mathrm{FN}_g(c)$$

$$= \#^* \mathcal{E}_m(\hat{g}) - \lambda_3 \sum_c \mathrm{FN}_g(c) - \sum_c \mathrm{FN}_g(c)$$

$$= \#^* \mathcal{E}_m(\hat{g}) - (1 + \lambda_3)\mathrm{FNE}(\hat{g})\#^*$$

for some $\lambda_2 \in [0, 1]$, and since $\lambda_3 = \lambda_2 \frac{\sum_c \mathrm{CM}_g(c)}{\sum_c \mathrm{FN}_g(c)} \leq \lambda_2$, we have $\lambda_3 \in [0, 1]$. We thus have that

$$\mathrm{CE}(\hat{g}) = \#^* \frac{\mathcal{E}_m(\hat{g}) - (1 + \lambda_3)\mathrm{FNE}(\hat{g})\#^*}{\#^*(1 - \mathrm{FNE}(\hat{g}))}$$

$$= \frac{\mathcal{E}_m(\hat{g}) - (1 + \lambda_3)\mathrm{FNE}(\hat{g})}{1 - \mathrm{FNE}(\hat{g})}$$

### F.3 MAIN PROOF

**Forward direction** Assume $\mathcal{E}_m(\hat{g}) \leq \epsilon$. We have that

- We proceed by using the quantities from above, bounding FNE.

$$\mathrm{FNE}(\hat{g}) = \frac{\sum_c \mathrm{FN}_g(c)}{\#^*}$$

$$= \frac{\#^* \mathcal{E}_m(\hat{g}) - \lambda_2 \sum_c \mathrm{CM}_g(c) - \min_\tau \mathbb{E}[|\mathrm{conf}_\tau(\hat{g}(x), g^*(x))|]}{\#^*}$$

$$\leq \mathcal{E}_m(\hat{g})$$

$$\leq \epsilon$$

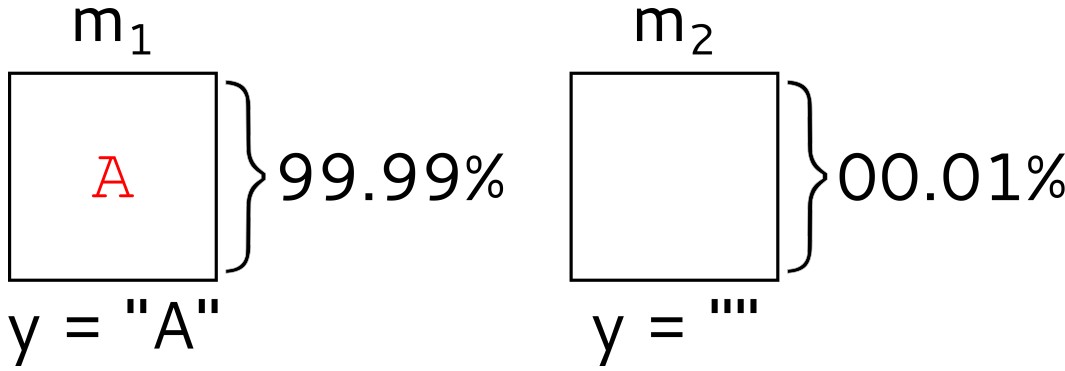

Figure 7: Counterexample motivating the $q(m_2) \leq P_m(m_2)$ requirement

- Since we know from Section E.2.1 that $\text{FP}_g \leq \sum_c \text{FN}_g(c)$ we have

$$\text{FPE}(\hat{g}) = \frac{\text{FP}_g}{\#^*}$$
$$\leq \frac{\sum_c \text{FN}_g(c)}{\#^*}$$
$$\leq \epsilon$$

- If $\epsilon \geq \frac{1}{2}$ then clearly $\text{CE}(\hat{g}) \leq 1 = 2\epsilon$. If $\epsilon < \frac{1}{2}$ we have that $1 - \text{FNE}(\hat{g}) > \frac{1}{2}$ and thus

$$\text{CE}(\hat{g}) = \frac{\mathcal{E}_m(\hat{g}) - (1 + \lambda_3)\text{FNE}(\hat{g})}{1 - \text{FNE}(\hat{g})}$$
$$\leq 2(\mathcal{E}_m(\hat{g}) - (1 + \lambda_3)\text{FNE}(\hat{g}))$$
$$\leq 2\mathcal{E}_m(\hat{g})$$
$$\leq 2\epsilon$$

as desired

**Backward Direction** Assume that $\text{CE}(\hat{g}) \leq \frac{1}{2}\epsilon$ and $\text{FNE}(\hat{g}) \leq \frac{1}{4}\epsilon$. We then have that

$$\mathcal{E}_m = \text{CE}(\hat{g})(1 - \text{FNE}(\hat{g})) + (1 + \lambda_3)\text{FNE}(\hat{g})$$
$$\leq \text{CE}(\hat{g}) + 2\text{FNE}(\hat{g})$$
$$\leq \epsilon$$

## G    COUNTEREXAMPLES FOR LESS INTUITIVE ASSUMPTIONS

### G.1    TRANSLATIONAL INVARIANCE OF BACKGROUND DISTRIBUTION

We assume that $P_b$ is translationally invariant. For an example of what happens if this assumption is broken, consider a version of DIGITCIRCLE where one digit always appears on the left side of the image, and is not read as part of the output $y^*$. While one might think this would lead to the possibility of high motif error without high end-to-end error, the locality of $\hat{g}$ ensures that the motif is predicted correctly despite not being used, as $\hat{g}$ does not "know" that the motif will not be useful. However, if $P_b$ were not translationally invariant, it would be possible for e.g., the background to be systematically darker on the left side of the image, with the motif prediction being slightly off center to take this into account and not report the motif if it is in the darker region. This would not affect end-to-end error but would affect motif error.

### G.2    NO INCREASE IN PROBABILITY MASS FOR PERTURBATIONS

To demonstrate the necessity of the $q(m_2) \leq P_m(m_2)$ requirement, consider the domain shown in Figure 7, which has exactly $M = \{m_1, m_2\}$ with $m_1$ having $1 - \iota$ probability (depicted is

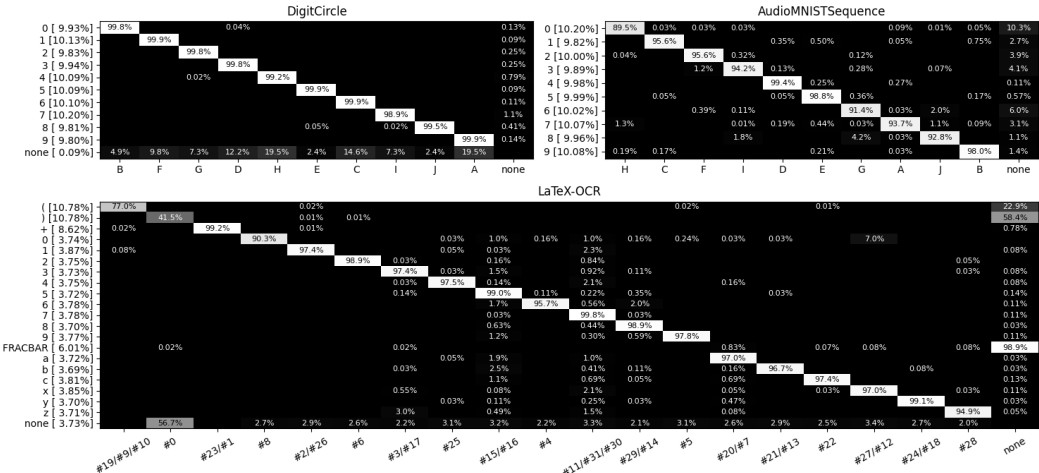

Figure 8: Confusion Matrix of 10k unseen samples computed for seed=1 across all domains. Each row represents a true motif being recognized and column represents a channel in the model's motif output. False positive and false negative motifs are placed into the `none` rows and columns, respectively. Each row is labeled by the percentage of motifs falling into the row, and each row's cells are then normalized to add to 1. We then permute to align along the diagonal. For LaTeX-OCR, we use more channels than there are symbol types so we merge channels together for display and analysis.

$\iota = 0.01\%$). Clearly, this domain trivially satisfies NON-OVERLAPPING and MOTIF-SUFFICIENCY as there is only one motif. Additionally, if we let $\psi(m_2|m_1) = 1$ and $\psi(\perp|m_2) = 1$, we have that $\psi$ clearly satisfies the support requirement and since $q(m_2) = 1 - \iota$, we have that $\sum_m q(m) = 1 - \iota$ so this domain satisfies the $\alpha$-MOTIF-NECESSITY assumption with $\alpha = 1 - \iota$. However, we have that we can set $\hat{g}(x) = 0$ and $\hat{h}(x) = $ "A", giving $\mathcal{E}(\hat{h} \circ \hat{g}) = \iota$ and $\mathcal{E}_m(\hat{g}) = 1$. This clearly breaks our proof since we can make $\iota$ arbitrarily small while not changing $\alpha$ much as it converges to 1.

## H  CONFUSION

Figure 8 depicts appropriately permuted confusion matrices for each domain. Our model generally assigns each true motif to a channel or set of channels in the sparse layer. The main exception is that in LaTeX-OCR, the fraction bar is never recognized, and `()` are only sometimes recognized. In other seeds, + exhibits similar behavior to `()`.

## I  SPARSITY AS AN INFORMATION BOUND

### I.1  CONNECTION TO INFORMATION BOUND

Sparsity induces an information bound by limiting the amount of information in the intermediate representation. Specifically, if we let $\mathcal{X}$ be a random variable for the input, and $\mathcal{M} = g^*(\mathcal{X})$ be the motif layer, we have that we can bound the mutual information between inputs and motifs as $I(\mathcal{X}, \mathcal{M}) \leq H(\mathcal{M})$, where $H(\cdot)$ is entropy. Thus, to bound mutual information, it is sufficient to bound $H(\mathcal{M})$. We first can break it into per-channel components: $H(\mathcal{M}) \leq \sum_{i,c} H(\mathcal{M}[i,c])$, Then, let $\delta_{i,c}$ denote the density of channel $c$ at position $i$, and $\eta \geq H(\mathcal{M}[i,c]|\mathcal{M}[i,c] \neq 0)$ be a bound on the amount of entropy in each nonzero activation (see Appendix I.2). Then we apply the chain rule to get $H(\mathcal{M}[i,c]) \leq H(B(\delta_{i,c})) + \eta\delta_{i,c}$ where $B(\cdot)$ is the Bernoulli distribution. Thus, $H(\mathcal{M}) \leq \sum_{i,c} H(B(\delta_{i,c})) + Sn\eta\delta$, where $S$ is the size of the image in pixels and $n$ is the number of channels, and $\delta$ is defined as in section 2. Finally, using

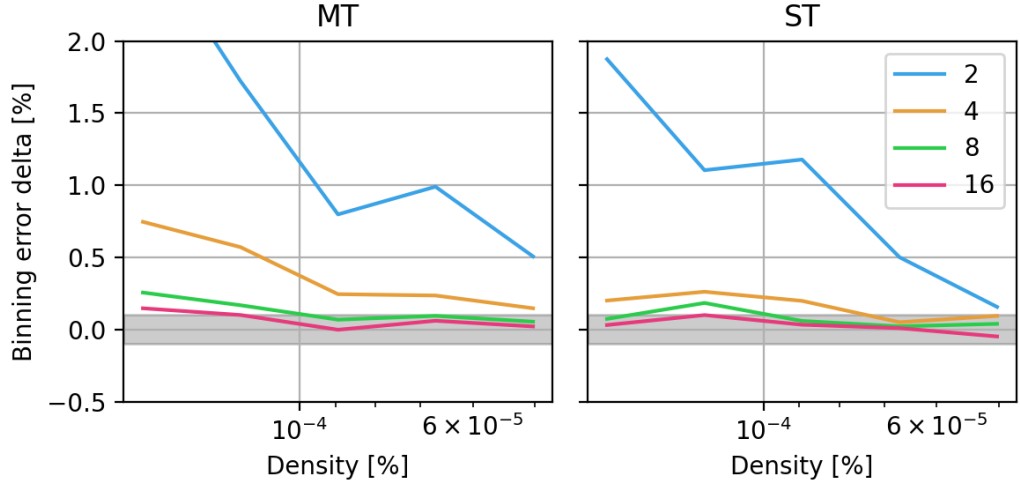

Figure 9: Increase in error when binning. Each series represents a different bin count, as annotated in the legend. Density is log-scaled and reversed to indicate training progress. MT is the model tracked in the rest of the paper, ST is the model as defined in Appendix K.3

Jensen's inequality (as $H(B(t))$ is concave):

$$I(\mathcal{X}, \mathcal{M}) \leq H(\mathcal{M}) \leq Sn(H(B(\delta)) + \eta\delta).$$

Since the computed bound is a monotonic function in $\delta$, where as $\delta \to 0$, the bound approaches 0, we can see that a sparsity bound can be used as an informational bottleneck for any information bound of a user's choosing.

## I.2 Entropy upper bound

To compute our entropy upper bound, we must first compute $\eta$, as defined in Section I.1. To compute this, we bin the nonzero activations into $2^k$ bins by quantile. We set $\eta$ to be the smallest value of $k$ that does not substantially affect the accuracy of the model (we consider 0.1% to be a reasonable threshold for this purpose). Figure 9 shows the result of this experiment, averaged across 9 seeds. The general downward trend in error caused by binning as density decreases demonstrates that reducing the number of motifs reduces the importance of the precise magnitudes. For the purposes of entropy bounding, we can use $\eta = \log(16) = 4b$.

## J Predicting motif error.

Figure 10 shows the relationship between the motif errors and the overall end-to-end error for DIGITCIRCLE. There is no relationship for FPE, but there is a positive relationship for CE, implying that a strategy where one trains several models and then chooses the one with the best validation error is a good way to reduce CE and thereby improve motif quality. This provides further evidence for Motif Identifiability (though the primary evidence for this remains that this model is able to achieve low FPE and CE in general, as training itself focuses on reducing end-to-end error via the loss function). While this may seem to contradict the result in Section 5.2, it in fact does not. Within a single model, tightening the density has inverse effects on end-to-end error and CE, but separately, some models are in general more or less accurate.

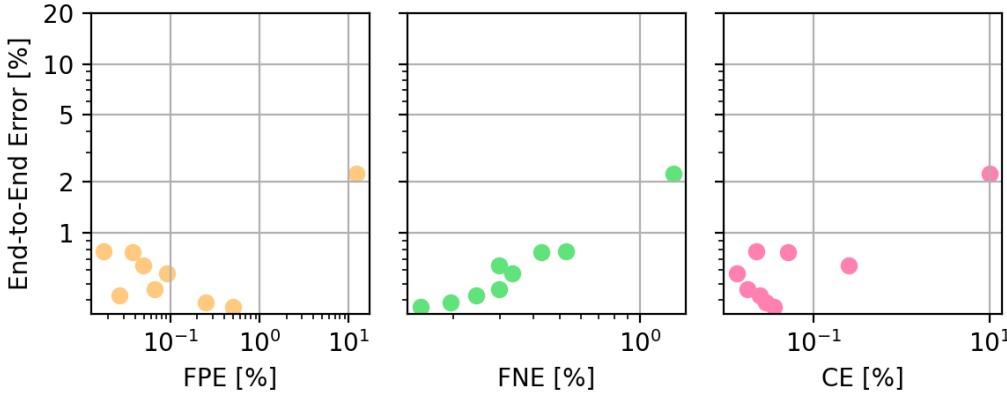

Figure 10: Model error versus FPE and CE, at $1.1\times$ the minimum sparsity. All are log-scaled to highlight the low-error region. Each dot represents a single model training seed.

## K COMPARISONS BETWEEN SPARLING AND OTHER TECHNIQUES FOR SPARSITY

In this section we compare to alternatives of the SPARLING model. For all comparisons, we keep the model architecture fixed and only modify the Sparse layer.

### K.1 BASELINES

We consider two other approaches to ensuring the creation of sparse motifs, both taking the form of auxiliary regularization losses. In both cases, we vary loss weight to analyze how that affects error and sparsity. First, we consider $L_1$ loss. In our implementation, we use an affine batch normalization layer followed by a ReLU. The output of the ReLU is then used in an auxiliary $L_1$ loss[12]. This approach is discussed in Jiang et al. (2015). We also consider using $KL$-divergence loss as in Jiang et al. (2015). The approach is to apply a sigmoid, then compute a KL-divergence between the Bernoulli implied by the mean activation of the sigmoid and a target sparsity value (we use 99.995% to perform a direct comparison). While this usually is done across the training data (Ng, 2011), we instead enforce the loss across all positions and channels, but per-batch (the mean sparsity should be similar in each batch). Our other modification, in order to induce true sparsity, is to, after the sigmoid layer (where the loss is computed), subtract 0.5 and apply a ReLU layer.

Table 1 shows the results of using $L_1$ as a method for encouraging sparsity. There are two weight regimes, where when $\lambda \leq 1$, we end up with high density (relative to the theoretical minimum) but low error, and when $\lambda \geq 2$, we end up with high-error model. Even in the latter case, the $L_1$ loss does not consistently push density down to the level of SPARLING, suggesting it might be insufficiently strong as a learning signal. In our experiments, the $KL$-divergence was unable to achieve a density below 0.1%, even when we used a loss weight as high as $\lambda = 10^5$ and $3 \times 10^6$ steps (much more than was necessary for convergence of the $L_1$ model). Thus, we conclude that it is unsuitable for encouraging the kind of sparsity we are interested in.

### K.2 ABLATIONS

We consider two ablations: First, is the batch normalization we place before our sparse layer necessary? Second, is the adaptive sparsity algorithm we use necessary? These ablations are only evaluated on DIGITCIRCLE as it is the domain where simpler techniques would work best.

We find that including a batch normalization before the sparsity layer is crucial. Without a batch normalization layer, over 9 runs, the best model gets an E2EE of 71% (the best ST model gets an

---

[12]This approach parameterizes the same model class as SPARLING; both act as a ReLU in a forward pass

| | $L_1$ | | | | | SPARLING |
|---|---|---|---|---|---|---|
| | $\lambda = 0.1$ | $\lambda = 1$ | $\lambda = 2$ | $\lambda = 5$ | $\lambda = 10$ | MT |
| FPE [%] | 99.99 | 99.90 | 91.25 | 95.99 | 97.63 | 1.48 [0.07-4.23] |
| FNE [%] | 0.00 | 0.00 | 58.09 | 73.12 | 84.51 | 0.42 [0.25-0.67] |
| CE [%] | 50.34 | 47.84 | 45.65 | 50.85 | 33.82 | 1.16 [0.03-3.39] |
| E2EE [%] | 0.68 | 2.85 | 70.31 | 75.00 | 73.20 | 0.74 [0.47-1.15] |
| Density [%] | 37 | 4.6 | 0.021 | 0.031 | 0.028 | 0.005 |

Table 1: Results of $L_1$ experiment on DIGITCIRCLE. As $L_1$ increases, the density decreases, but end-to-end error becomes $> 50\%$, and CE/FPE never improve to the level of SPARLING. SPARLING is able to keep error low while achieving lower density than $L_1$ with any $\lambda$ value we tried.

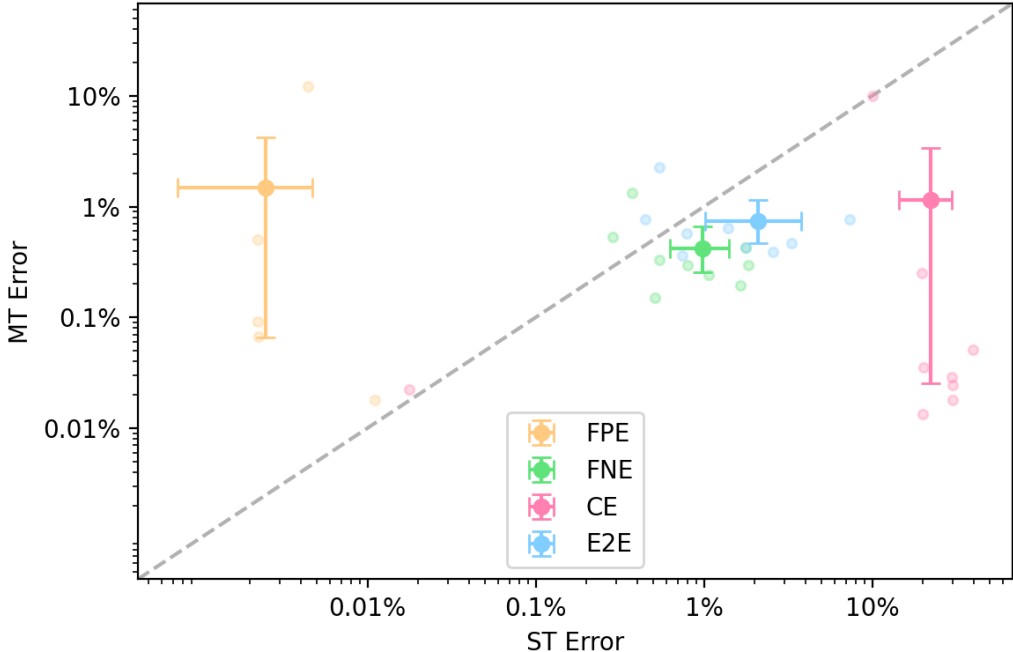

Figure 11: SPARLING using MT (as in the main figures) vs ST

error of 52.48%); in essence, it is not able to learn the task at all. Additionally, annealing (Algorithm 1) is clearly necessary: when started with the annealing algorithm's final and penultimate $\delta$ values, the model converged to E2EE values of 68% and 71% respectively.

### K.3 SINGLE THRESHOLD

In this section, we consider a variation to the quantile function. We call this the *single threshold (ST)* sparsity approach, as opposed to the *multiple thresholds (MT)* technique described in Section 4, where we take the quantile across the entire input (batch axis, dimensional axes, channel axis). In this case, the channels can have differing resulting densities that average together to the target $\delta$. More precisely, we use the quantile function $q_{\text{ST}} : \mathbb{R}^{B \times d_1 \times \dots \times d_k \times n} \times \mathbb{R} \to \mathbb{R}$, implemented such that

$$p \approx \frac{1}{BSC} \sum_{b,i,c} \mathbf{1}(z[b,i,c] \leq q_{\text{ST}}(z,p)).$$

As seen in Figure 11, ST performs substantially worse in terms of CE and E2EE, while performing better with respect to FPE. Without the constraint that the motifs have equivalent density across

each channel, some motifs are being used to represent multiple digits, which substantially increases confusion error, but also reduces false positives. In general, the MT model is superior as it has reasonable FPE and substantially lower CE/E2EE.

## L    COMPARISON TO DIRECTLY LEARNING THE MOTIFS

|  | SPARLING [mean] | DIRECT [mean] | Ratio [of means] |
|---|---|---|---|
| DigitCircle | 1.24 | 0.01 | 0.01 |
| LaTeX-OCR | 6.55 | 0.12 | 0.02 |
| LaTeX-OCR [without +()] | 2.96 | 0.10 | 0.03 |
| AudioMNISTSequence/train | 5.41 | 0.61 | 0.11 |
| AudioMNISTSequence/test | 8.01 | 4.28 | 0.53 |

Table 2: Error [%] and ratios between errors. All are computed as a mean across 9 seeds

The purpose of SPARLING is to be able to learn intermediate state without having to have access to any training data on the intermediate state. In this section, we analyze how well it does at this goal, by comparing it to DIRECT, a setting where we train and evaluate on the intermediate state directly. Specifically, we construct datasets for each task of single motifs and train and test models on these datasets, then also test SPARLING on these datasets.

In the case of DIGITCIRCLE and LATEX-OCR, DIRECT is a trivial task as there is no distributional shift in the motif samples used to train and evaluate the model – effectively, DIRECT is tested on the training set. Thus, DIRECT gets ∼0% error.

However, on the AUDIOMNISTSEQUENCE task, the DIRECT has non-negligible error, with 0.61% error on the training sample distribution but a much higher 4.28% error on the testing sample distribution. Meanwhile, SPARLING increases substantially less, from 5.41% to 8.01%. This is because the error in SPARLING comes from two sources: the underlying uncertainty in prediction it shares with the DIRECT technique, and epistemic uncertainty related to the problem of identifying motifs from end-to-end data. This latter error evidently does not scale linearly with the difficulty of the underlying task.

## M    SPLICING DOMAIN

We also consider the original splicing domain, hypothesizing that on a domain that does not satisfy our assumptions in Section 3.3, SPARLING will not perform well but can perform better than chance. To keep things simple, we use the Jaganathan et al. (2019) architecture as the $\hat{h}$ model and a simple convolutional stack identical to the adjustment model from Gupta et al. (2024) as the $\hat{g}$ model. To ensure our experiment is picking up on a real signal, we will exclude the local splice site motifs (LSSI sites) from the set of true motifs for the purposes of analysis, as these sites can be found trivially from the end-to-end data, instead, we only evaluate on the other protein binding sites.

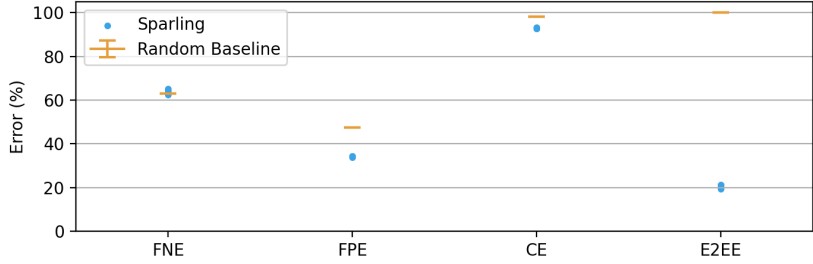

Figure 12: Results on the splicing domain. Results are presented per error metric for both 4 runs of SPARLING and 95% CI of a boostrap mean of 10 runs of a matched randomized baseline.

SPARLING achieves reasonable end-to-end performance, but does not perform as well as the other three domains on motif prediction (see Figure 12). However, we find that it consistently outperforms a random chance baseline in the most important error metric, CE—indicating that it is correctly classifying motifs. The other error metrics are more mixed, while it outperforms the baseline in FPE, it underperforms it in FNE suggesting that the model is producing duplicate activations, which leads to insufficient coverage of the motifs. Overall, this is consistent with our hypothesis that while Motif Identifiability is only possible given certain assumptions, SPARLING is capable of picking up some signal even when these assumptions are not met.

# N  REPRODUCIBILITY

## N.1  REPRODUCIBILITY REPOSITORY

A repository that will allow you to reproduce the results in this paper can be found at `https://github.com/kavigupta/sparling-repro`.

## N.2  COMPUTE USAGE

All our experiments were performed on NVIDIA GeForce GTX 1080 Ti (12GiB VRAM) or Quadro RTX 5000 (16GiB VRAM) GPUs. On average, DIGITCIRCLE experiments took 4 hours each to train, LATEX-OCR experiments took 14 days each to train, and AUDIOMNISTSEQUENCE experiments took 5 days to train. In total, we used about 350 GPU days of compute for the experiments reported in the body of the paper, 250 GPU days for the experiments referenced in footnotes/the appendix, and 200 GPU days of compute for exploratory experiments that were not referenced.

