# OpenReview forum: "Sparling: End-to-End Spatial Concept Learning via Extremely Sparse Activations"
_ICLR.cc/2026/Conference — ICLR 2026 Poster_

### Official Review · Reviewer_sThB · 2025-10-19

**Soundness:** 3
**Presentation:** 2
**Contribution:** 2
**Rating:** 2
**Confidence:** 3

**Summary:**

The authors propose an end-to-end spatial concept learning framework that identifies interpretable intermediate representations (called motifs) through extremely sparse activations. The paper introduces a Motif Identifiability Theorem, proving that under certain assumptions (locality, sparsity, and independence), true latent motifs can be recovered solely from end-to-end supervision. They further present the adaptive sparsity algorithm, which gradually enforces ultra-sparse activations during training to achieve interpretability without supervision. Experiments on synthetic datasets  show that SPARLING can localize intermediate states with over 90% accuracy, bridging theoretical guarantees and practical learning.

**Strengths:**

The paper establishes a clear theoretical foundation with the Motif Identifiability Theorem (Section 3), offering formal conditions where sparse, local latent variables are identifiable. This bridges a long-standing gap between interpretability and end-to-end deep learning by providing provable guarantees rather than heuristic explanations.

From an engineering perspective, the Spatial Sparsity Layer and Adaptive Sparsity Algorithm (Section 4) are elegant and practical. They show how to gradually enforce 99%+ sparsity while keeping models trainable, making the framework useful beyond the synthetic domains tested.

**Weaknesses:**

The experiments mainly rely on highly synthetic datasets—DIGITCIRCLE, LATEX-OCR, and AUDIOMNISTSEQUENCE (Section 5.1). This leaves uncertainty about whether SPARLING’s assumptions (e.g., NON-OVERLAPPING or PATCH-INDEPENDENCE) hold in real tasks such as natural images or genomics data.

The theorem depends on strong constraints (Section 3.3 – lines 269–324) like NON-OVERLAPPING motifs and PATCH-INDEPENDENCE, which may not be satisfied in practical domains. For example, line 270 explicitly admits that “this assumption… excludes some of our domains because of how large the distance needs to be.”

Figure 5 and the discussion in lines 432–480 show SPARLING’s end-to-end error is higher than non-sparse baselines, but the explanation remains speculative (“we theorize that this is because our constraint… requires the model to ‘commit’”). There is also no direct quantitative comparison with modern interpretability methods such as saliency or concept bottleneck models.

**Questions:**

How sensitive is SPARLING’s performance to the chosen sparsity target (e.g., 99% vs 99.9%)—does extreme sparsity always help motif identifiability?

Can the authors clarify whether the Motif Identifiability Theorem can be extended to overlapping or correlated motifs, and if so, what modifications would be needed to the current assumptions?

---

> ### Author Response · Authors · 2025-11-19
>
> Thank you for your helpful review! Below are our responses to your questions, split into two parts for character count reasons.
>
> > The experiments mainly rely on highly synthetic datasets—DIGITCIRCLE, LATEX-OCR, and AUDIOMNISTSEQUENCE (Section 5.1). This leaves uncertainty about whether SPARLING’s assumptions (e.g., NON-OVERLAPPING or PATCH-INDEPENDENCE) hold in real tasks such as natural images or genomics data.
>
> There is a long tradition in Machine Learning of using simplified forms of more complex real-world algorithms to gain theoretical insights in how the more complex algorithms work. We see our paper as an instance of this tendency. In particular, there is a recent result in genomics that shows a more complicated form of this algorithm being used to infer difficult-to-observe intermediate values in a biological process. In this paper, we look at a simplification of that original algorithm and study the conditions under which it is possible to make these kinds of inferences, in the absence of any data about the intermediate whatsoever.
>
> Remarkably, this simplified algorithm actually works on nontrivial domains, showing that we did not oversimplify for the sake of theoretical analysis. This simplified algorithm is not sufficiently powerful to fully capture this genomic domain (though it does identify the latent at a rate better than chance), but does provide insights into why discovering these intermediate values is at all possible, and we hope will serve as a first step towards developing even better algorithms for automatic latent identification using neural networks.
>
> > The theorem depends on strong constraints (Section 3.3 – lines 269–324) like NON-OVERLAPPING motifs and PATCH-INDEPENDENCE, which may not be satisfied in practical domains. For example, line 270 explicitly admits that “this assumption… excludes some of our domains because of how large the distance needs to be.”
> > Can the authors clarify whether the Motif Identifiability Theorem can be extended to overlapping or correlated motifs, and if so, what modifications would be needed to the current assumptions?
>
> While we acknowledge that the assumptions necessary to prove the theorem are a bit more stringent than apply to many domains, we believe that this is reasonable as there are loosened versions of the assumptions still make sense.
>
> The assumption regarding non-overlap arises because we make very limited assumptions about structure of the individual motifs. Specifically, we only assume that they follow some distribution and are constrained to a bounding box. We would be able to relax the assumption of non-overlapping if we added some assumptions about how much of a motif’s footprint is needed in order to make a high confidence prediction about the presence of the motif. Because our assumption is so weak, we have to account for the possibility that a local model with access to the corners of two motifs could fully determine what each of these motifs are, and then use that information to selectively not report one of the motifs (e.g., not reporting a u that appears after a q in English text). In practical domains, this is a relatively rare edge case; but excluding it without excluding overlap in general requires significantly more complicated assumptions.
>
> In practice, LaTeX-OCR does not meet the requirements of non-overlapping, but the algorithm still works, demonstrating that in practice the algorithm is robust to this assumption not being satisfied, as long as the edge case described above is not encountered.

---

> > ### Author Response · Authors · 2025-11-19
> >
> > > Figure 5 and the discussion in lines 432–480 show SPARLING’s end-to-end error is higher than non-sparse baselines, but the explanation remains speculative (“we theorize that this is because our constraint… requires the model to ‘commit’”).
> >
> > This is a hypothesis that is then validated by the Retrained setting, which demonstrates that the motifs, absent the bottleneck, provide sufficient information to match the Non-Sparse model.
> >
> > As the motifs without the bottleneck are directionally correct as a summary of the image in a way that can predict the output, this demonstrates that the concern is with the bottleneck itself, rather than the motif models. We have updated our language to make this more clear.
> >
> > > There is also no direct quantitative comparison with modern interpretability methods such as saliency or concept bottleneck models.
> >
> > We are not attempting to solve the same problem as saliency; as saliency mapping produces local explanations, whereas we produce global ones. In other words, our motifs layer M produces a listing of all information in the input that is being used to predict the output, whereas a saliency map produces a listing of information that, if locally changed, would alter the output. The difference is that e.g., if the output is true if either features A or B are true, and both features appear in a scene where feature A is stronger than B, then feature B will not be emphasized in the saliency map, whereas in Sparling, both will be present.
> >
> > On the other hand, we do not doubt that concept bottlenecks would be able to perform as well as we do on these tasks *if given intermediate layer supervision* as required by the concept bottleneck algorithm. Our improvement over concept bottlenecks is qualitative in that we do not require any supervision on the intermediate layer, and as such a quantitative comparison would not make sense.
> >
> > > How sensitive is SPARLING’s performance to the chosen sparsity target (e.g., 99% vs 99.9%)—does extreme sparsity always help motif identifiability?
> >
> > In Figure 4, we see generally monotonic reductions in motif error as sparsity increases towards the critical 1 - delta* value; having a density 10x  higher than the critical density value (as in your 99% vs 99.9% example) generally results in an unacceptably high confusion error, and fairly high false positive error. Confusion error is the most significant metric, as it is what allows us to assign semantic meaning to individual channels in the motif model output. False negative error generally has the opposite pattern, though usually stays low (LaTeX-OCR is the exception because alpha-Motif-Importance is not satisfied for parentheses and the + operator, which can be inferred from context due to the lack of multiple types of parentheses and operators). In general, having a sparsity that is too low results in motif identifiability not happening.

---

### Official Review · Reviewer_g8Sy · 2025-10-22

**Soundness:** 4
**Presentation:** 3
**Contribution:** 4
**Rating:** 6
**Confidence:** 3

**Summary:**

The paper introduces the Motif Identifiability Theorem, which proves that under conditions of locality, extreme sparsity, and certain distributional assumptions, a model with low end-to-end error must also accurately recover the true motifs if its bottleneck matches the real motif density. Building on this theory, the authors propose SPARLING, a training method that enforces highly sparse intermediate activations to learn spatial motifs directly from data. Using a quantile-based spatial sparsity layer with annealed target density and per-channel thresholds, SPARLING achieves over 90 percent correct motif localization on three semi-synthetic datasets: DIGITCIRCLE, LATEX-OCR, and AUDIOMNISTSEQUENCE.

**Strengths:**

- Without direct supervision of intermediate motifs, the proposed method achieves interpretable and accurate motifs while maintaining competitive end-to-end performance across three evaluated tasks.
- The work shows interesting trade-offs with their empirical results. As shown in figure 4, increasing sparsity reduces the confusion error (CE) and motif channels become better separated. However, it slightly increases end-to-end task error (E2EE), given that model's information bottleneck is tighter.
- As shown in appendix K.1, the work compares Sparling with L1 regularization as a baseline to encourage sparsity. Although by using a higher $\lambda$ one can decrease the density using L1 regularization, the E2EE significantly increases. Using Sparling however, the work shows that it can achieve density lower than $\lambda =10$ of L1 regularization but with E2EE comparable with $\lambda=0.1$.
- The work tests the proposed method on domains that violates the assumptions. As shown in appendix M, in splicing domain that doesn't met the assumptions, although CE is high, it shows that it outperforms random chance baseline, highlighting that the method is able to capture meaningful signals even when its assumptions are not satisfied.

**Weaknesses:**

-  All the core results are evaluated on synthetic or synthetic-ized tasks (digits in circles, LaTeX OCR with controlled rendering, spoken digits). The work would be much stronger with a real-world dataset. Tasks like identifying small-objects such as ships on open water from satellite images. It would be interesting to see end-to-end training from scene labels such as vessels being present could discover localized motifs of where they are without providing bounding boxes.
 - As stated in Appendix N, approximately 350 GPU days were required for the main experiments. This is a substantial computational cost considering the relatively modest problem sizes, which limits the method’s practicality for larger-scale applications.

**Questions:**

- Many scenes have structured, non-stationary backgrounds. Could data augmentation (random crops/phase shifts) approximate patch-independence enough to recover motifs?

---

> ### Author Response · Authors · 2025-11-19
>
> Thank you for your helpful review! Below are our responses to your questions:
>
> > All the core results are evaluated on synthetic or synthetic-ized tasks (digits in circles, LaTeX OCR with controlled rendering, spoken digits). The work would be much stronger with a real-world dataset. Tasks like identifying small-objects such as ships on open water from satellite images. It would be interesting to see end-to-end training from scene labels such as vessels being present could discover localized motifs of where they are without providing bounding boxes.
>
> There is a long tradition in Machine Learning of using simplified forms of more complex real-world algorithms to gain theoretical insights in how the more complex algorithms work. We see our paper as an instance of this tendency. In particular, there is a recent result in genomics that shows a more complicated form of this algorithm being used to infer difficult-to-observe intermediate values in a biological process. In this paper, we look at a simplification of that original algorithm and study the conditions under which it is possible to make these kinds of inferences, in the absence of any data about the intermediate whatsoever.
>
> Remarkably, this simplified algorithm actually works on nontrivial domains, showing that we did not oversimplify for the sake of theoretical analysis. This simplified algorithm is not sufficiently powerful to fully capture this genomic domain (though it does identify the latent at a rate better than chance), but does provide insights into why discovering these intermediate values is at all possible, and we hope will serve as a first step towards developing even better algorithms for automatic latent identification using neural networks.
>
> > As stated in Appendix N, approximately 350 GPU days were required for the main experiments. This is a substantial computational cost considering the relatively modest problem sizes, which limits the method’s practicality for larger-scale applications.
>
> In practice we could probably achieve similar results with a much faster annealing schedule or implement low-level optimizations to better utilize GPU resources. As this paper focused on whether or not motif identification is possible, we did not spend any time optimizing this algorithm.
>
> Additionally, the 350 days also reflects the time spent to run 9 seeds of each experiment on relatively archaic hardware (2017-2018 era GPUs: a GTX 1080 Ti and a RTX 5000), and should not be treated as the time taken to run the algorithm, even unmodified, on modern hardware. The longest time taken by a single experiment was about 2 weeks on a single GPU with this hardware, which remains within reach for most labs.
>
> > Many scenes have structured, non-stationary backgrounds. Could data augmentation (random crops/phase shifts) approximate patch-independence enough to recover motifs?
>
> Yes; this should provide the translational independence needed, unless there is a correlation between the motifs and the background that cannot be simply treated as “part of the motif” (e..g., if in a typewriter OCR task, there is always some typewriter inkblot within the loop of an a, that can be treated as part of the “a” motif; but if there is always a streak across the page whenever there’s an a, that cannot.)

---

> > ### Comment · Reviewer_g8Sy · 2025-11-26
> >
> > Thank you to the authors for their response. Sparling indeed offers a useful perspective on recovering motifs when its theoretical conditions are met. However, as the paper itself acknowledges, these assumptions rarely hold for real-world data, which limits practical applicability. In addition, given that training on a modern-sized problem currently requires roughly two weeks, the method appears computationally infeasible for large-scale settings. I will therefore keep my original score.

---

### Official Review · Reviewer_uhfz · 2025-10-29

**Soundness:** 3
**Presentation:** 2
**Contribution:** 3
**Rating:** 6
**Confidence:** 3

**Summary:**

This paper theoretically proves the motifs (or latent variables) of a task can be accurately identified by a neural network that is trained end-to-end. Specifically, the paper proves an upper bound for the motif error when the end-to-end error is low under certain assumptions. The paper shows that this kind of motif identifiability can be achieved by enforcing extremely sparse activations in the training process and empirically validates the algorithm on several synthetic tasks including digit circle recognition, latex-ocr, and audio sequence identification.

**Strengths:**

1.	The scope of the paper is clear: it focuses on problems in which latent variables are sparse non-overlapping spatial positions. Based on this, the paper addresses an interesting problem: can a model automatically identify these latent variables in its hidden representation when trained end-to-end?

2.	The paper has good theoretical contribution: an identifiability result for local, extremely sparse spatial concepts learned end-to-end, under explicit assumptions. The bound that relates end-to-end error to motif error is valuable.

3.	Novel algorithm design: Besides the theoretical results, the paper proposes a practical algorithm to boost the sparsity of intermediate representations via per-channel quantile thresholds and annealing. This sparsity ratio is often unattainable by standard regularizers.

**Weaknesses:**

1.	The theorem requires $\delta(\hat{g})=\delta^\*$. However, in practice, $\delta^*$ might be unknown. There is no robustness analysis for misspecified $\delta$.

2.	Experiments remain synthetic (including LATEX-OCR). Real-world validations (e.g., natural scene text OCR, object detection) are absent.

**Questions:**

1.	Is there a relation between the identification of sparse spatial motifs and the superposition hypothesis[cite1] in mechanistic interpretability? I would appreciate a discussion on this comparison.

2.	What concrete modifications to the assumptions or the proof would allow overlapping motifs (common in detection with occlusion) or spatially correlated backgrounds? Are there any partial thoughts or empirical evidence with controlled overlap?

3.	How sensitive is SPARLING to $\delta$ misspecification (as stated in Weaknesses 1)?

---

> ### Author Response · Authors · 2025-11-19
>
> Thank you for your helpful review! We have placed our responses in two comments, for character count limitation reasons.
>
> > The theorem requires delta* = delta(\hat g). However, in practice, delta* might be unknown. There is no robustness analysis for misspecified delta.
>
> See Figure 4 for an robustness analysis on delta values that are too high: for the domains we consider, overestimating delta by more than about 2x makes motif error unacceptably high.
>
> Having too high of a delta value invalidates the theorem, as it means you can have multiple activations for a given motif, which could allow you to pathologically encode, e.g., the digits 1-3 as 010, 100, 110. This completely gets around the requirement that each motif be placed in its own channel. Too low a delta value means the task is impossible, as you cannot represent all the necessary information.
>
> This is part of the motivation for the annealing algorithm, in addition to it being necessary for the optimization problem to be solvable; in practice, you can decide that your delta is correct when the validation accuracy drops substantially (as you now are in the low-accuracy delta(\hat g) < delta* regime). We have added a reference to the annealing algorithm to the section where we introduce the delta(\hat g) = delta* assumption.
>
> > Experiments remain synthetic (including LATEX-OCR). Real-world validations (e.g., natural scene text OCR, object detection) are absent.
>
> There is a long tradition in Machine Learning of using simplified forms of more complex real-world algorithms to gain theoretical insights in how the more complex algorithms work. We see our paper as an instance of this tendency. In particular, there is a recent result in genomics that shows a more complicated form of this algorithm being used to infer difficult-to-observe intermediate values in a biological process. In this paper, we look at a simplification of that original algorithm and study the conditions under which it is possible to make these kinds of inferences, in the absence of any data about the intermediate whatsoever.
>
> Remarkably, this simplified algorithm actually works on nontrivial domains, showing that we did not oversimplify for the sake of theoretical analysis. This simplified algorithm is not sufficiently powerful to fully capture this genomic domain (though it does identify the latent at a rate better than chance), but does provide insights into why discovering these intermediate values is at all possible, and we hope will serve as a first step towards developing even better algorithms for automatic latent identification using neural networks.

---

> > ### Author Response · Authors · 2025-11-19
> >
> > > Is there a relation between the identification of sparse spatial motifs and the superposition hypothesis[cite1] in mechanistic interpretability?
> >
> > I assume you refer to superposition as it relates to polysemantic neurons, where multiple signals are passed through single neurons via linear superposition of nonlinear signals. The purpose of several of the assumptions (specifically locality, Non-Overlapping and Patch Independence) is in attempting to reduce the potential for such polysemantic neurons developing; specifically by isolating each potential motif to the point where it cannot be easily mixed with other motifs before the spatial sparsity layer. We are able to exclude the possibility of such superposition; as there are not multiple signals to superpose. Our theorem demonstrates that under our assumptions regarding the data distribution, we can guarantee that such polysemantic superposition is not occurring.
> >
> > From the perspective of interpretability, not having polysemantic neurons is desirable as it allows for interpretation of individual neurons. Other techniques such as Concept Bottlenecks, also attempt to do this.
> >
> > > What concrete modifications to the assumptions or the proof would allow overlapping motifs (common in detection with occlusion) or spatially correlated backgrounds? Are there any partial thoughts or empirical evidence with controlled overlap?
> >
> > The LaTeX results constitute controlled overlap in an empirical setting, as the “motifs” (really the bounding boxes, which is what is relevant to the proof) end up overlapping, and we find that this is not an issue for the algorithm.
> >
> > The assumption regarding non-overlap arises because we make very limited assumptions about structure of the individual motifs. Specifically, we only assume that they follow some distribution and are constrained to a bounding box. We would be able to relax the assumption of non-overlapping if we added some assumptions about how much of a motif’s footprint is needed in order to make a high confidence prediction about the presence of the motif. Because our assumption is so weak, we have to account for the possibility that a local model with access to the corners of two motifs could fully determine what each of these motifs are, and then use that information to selectively not report one of the motifs (e.g., not reporting a u that appears after a q in English text). In practical domains, this is a relatively rare edge case; but excluding it without excluding overlap in general requires significantly more complicated assumptions.

---

> > > ### Comment · Reviewer_uhfz · 2025-11-22
> > >
> > > I would like to thank the authors for their response. I will keep my score.

---

### Official Review · Reviewer_YS9h · 2025-10-31

**Soundness:** 3
**Presentation:** 2
**Contribution:** 3
**Rating:** 6
**Confidence:** 2

**Summary:**

The paper proves that motif identification is solvable under three main assumptions. it motivates that understanding motifs can help improvements of deep learning models in many real-world tasks such as RNA modeling and Latex OCR. The theoretical proofs seems sound and experiments are supporting the main general proofs, and the assumptions of the paper seems realistic for the real-world tasks.

**Strengths:**

The paper has several strengths:

* **Addressing an interesting problem:** Although I am not deeply familiar with this line of research, the problem of recognizing and understanding motifs appears to be a very interesting and meaningful task. The paper effectively motivates this problem through real-world applications such as RNA modeling, which are indeed highly relevant today.

* **Sound assumptions:** The assumptions made in the paper, aside from the non-overlapping constraint, seem reasonable. The idea that most motifs are important is both intuitive and realistic for the majority of relevant tasks.

* **Theoretical proofs:** While I am not fully familiar with the related literature, the theoretical proofs appear to be well supported, with clear reasoning and detailed supplementary arguments.

* **Experimental setup:** The experiments are well designed and align closely with the paper’s mot

**Weaknesses:**

I have only some minor weaknesses on presentation and details of the model:
* **1)** In real-world tasks, it is unclear how the model distinguishes between motifs and noise. As suggested by the paper, the mechanism by which the model identifies this difference is not explicitly explained.

* **2)** The description of the neural architecture is largely missing. Apart from a single equation related to sparsity in the Methods section, there is no clear definition of the model’s architecture or how it is implemented for learning.

* **3)** Lack of visualizations: Figures 1 and 2 provide clear and informative illustrations of motifs, but it would be helpful to include visualizations showing whether the model actually “attends to” or decodes these motifs. Beyond reporting accuracies, some interpretable visualization or analysis could demonstrate that the model is indeed identifying motifs.

* **4)** The title suggests that sparse activations play a central role, yet the connection between sparsity, motif learning, and the main theorem is not clearly articulated. This link should be made more explicit in the presentation.

**Questions:**

My questions are provided in weakness section.

---

> ### Author Response · Authors · 2025-11-19
>
> Thank you for your helpful review! We have responses to your questions below:
>
> 1) In real-world tasks, it is unclear how the model distinguishes between motifs and noise. As suggested by the paper, the mechanism by which the model identifies this difference is not explicitly explained.
>
> Our theorem proves that if the model is capable of learning the overall function end-to-end, it must have learned how to distinguish the motifs from noise (and each other). The intuitive basis for why this works is that if $\hat g$ has not learned to distinguish motifs, due to the sparsity constraint, the $\hat h$ model will not have sufficient information to predict the output with high accuracy.
>
> This leaves the question of why models are able to learn end-to-end in the presence of such a sparsity constraint. In general, even without a sparsity constraint, models inevitably place more emphasis on parts of the input that are related to the task at hand. We exploit this via our adaptive sparsity approach, which gradually tightens the sparsity and forces the model to make decisions about what signals to preserve. (Batch normalization ends up being critical to preventing the model from getting “stuck” with valuable signals lost.)
>
> Sorry if we did not understand your question, we are happy to follow up with additional clarifications.
>
> 2) The description of the neural architecture is largely missing. Apart from a single equation related to sparsity in the Methods section, there is no clear definition of the model’s architecture or how it is implemented for learning.
>
> We discuss architecture not in the Methods section but in the Experiments section, in the “Architecture and training” paragraph. If you would think moving this to the Methods section would improve the organizational clarity of the paper, we can do so.
>
> 3) Lack of visualizations: Figures 1 and 2 provide clear and informative illustrations of motifs, but it would be helpful to include visualizations showing whether the model actually “attends to” or decodes these motifs. Beyond reporting accuracies, some interpretable visualization or analysis could demonstrate that the model is indeed identifying motifs.
>
> Thank you for your suggestion! We have added such an illustration to Appendix O, depicting the effect of perturbing individual motifs in a LaTeX-OCR context corresponding to a character on the output prediction. We feel this is somewhat more helpful in our context than a traditional saliency map as it shows how the motifs can be directly interpreted and manipulated. We also run a more systematic analysis, demonstrating that in general, these perturbations produce the expected effects in all domains.
>
> If you prefer, we can move this to the main content of the paper, alongside discussion of the motif locations in Figure 2 or alongside results in Figures 3-4.
>
> 4) The title suggests that sparse activations play a central role, yet the connection between sparsity, motif learning, and the main theorem is not clearly articulated. This link should be made more explicit in the presentation.
>
> The theorem requires extreme sparsity for the main result, in that it requires that delta(\hat g) = delta*; this in practice due to the non-overlapping assumption requires extremely high sparsity. We acknowledge that the use of language surrounding density rather than sparsity in the introduction of the theorem obscured this connection. A brief discussion of this connection has been added after the theorem statement in Section 3.1: “Note that $\delta(\hat g) = \delta^* $ enforces extreme sparsity, as $\delta^* $ is must be extremely small due to Non-Overlapping, and thus corresponds to an extreme sparsity setting; in practice $\delta^* $ is found via the adaptive sparsity algorithm.”
>
> Additionally, this property is empirically demonstrated in Figure 4, where it is clear that to reduce motif error, one must achieve extremely high sparsity. If you would like us to add additional discussion on the role of extreme sparsity to other sections, we can do so.

---

### Official Review · Reviewer_oPKW · 2025-11-03

**Soundness:** 1
**Presentation:** 1
**Contribution:** 1
**Rating:** 0
**Confidence:** 3

**Summary:**

This paper studies when and how semantically meaningful spatial concepts (called motifs) can be recovered purely from end-to-end supervision.

It introduces a Motif Identifiability Theorem, showing that under assumptions of locality, sparsity, non-overlap, patch independence, and α-motif-importance, one can identify the true latent motif map from low end-to-end error.

Building on this, the authors propose SPARLING, an algorithm that enforces extreme activation sparsity through an adaptive thresholding layer and annealed sparsity schedule.

Empirically, SPARLING is evaluated on synthetic and semi-realistic datasets—DIGITCIRCLE, LATEX-OCR, and AUDIOMNISTSEQUENCE—achieving accurate motif localization (> 90%) without direct supervision.

**Strengths:**

The paper provides a new theoretical formulation of identifiability for sparse local latent variables—a valuable bridge between statistical identifiability theory and deep concept-bottleneck learning. The emphasis on end-to-end identifiability (without explicit supervision) is conceptually novel.

**Weaknesses:**

1. The paper is not well-structured and well-written. The mathematical formalism is often dense and difficult to parse. Many places are hard to follow.

2. The theorem statements are long and self-referential but lack rigorous statement.

3. Figures are under-explained; axes and variables are often unlabeled.

**Questions:**

N/A

---

> ### Author Response · Authors · 2025-11-19
>
> > The paper is not well-structured and well-written. The mathematical formalism is often dense and difficult to parse. Many places are hard to follow.
>
> Thank you for your feedback; if you have any specific areas you would like us to improve, please let us know and we can make the necessary changes.
>
> > The theorem statements are long and self-referential but lack rigorous statement.
>
> We agree our theorem statements are long; this is typical for ML theory work. However, we believe that our theorems are not self-referential; if you could point us to self-reference we will eliminate it. The theorem has a rigorous statement in Appendix B (due to space constraints).
>
> > Figures are under-explained; axes and variables are often unlabeled.
>
> If you could point us to specific figures you would like additional explanation for, we can improve our explanations. As is, we believe all our axes and variables are labeled when appropriate. Figures 1 and 2 show images rather than plots, so do not need axis labels, and all points in Figure 2 are annotated in the legend, with further explanation in the caption. For Figure 3, the X axis is categorical and rather than being labeled in the axis is labeled using a color-coded legend (to match the colors used in Figure 4). In Figure 4 and 5, all axes and series are labeled. We also double-checked all figures in the appendix and they also have axis and variable labels when appropriate.

---

### Meta-Review · Area_Chair_Tr9D · 2026-01-13

**Summary:**

**Summary**
The paper establishes that motif identification is feasible under specific assumptions, emphasizing its potential to enhance deep learning applications like RNA modeling and Latex OCR. It introduces the Motif Identifiability Theorem, which asserts that true latent motifs can be identified from low end-to-end error if certain conditions are met. The authors present SPARLING, an algorithm designed to promote extreme activation sparsity, which is validated through experiments on synthetic datasets, achieving over 90% accuracy in motif localization. Overall, the research demonstrates that neural networks can effectively identify motifs through end-to-end training by enforcing sparse activations.

**Strengths**

- **Strong Theoretical Foundation**: The paper presents a well-supported theoretical contribution, including the Motif Identifiability Theorem, which establishes conditions for identifying sparse, local latent variables, bridging interpretability and end-to-end deep learning.

- **Innovative Algorithm Design**: It introduces a novel algorithm, the Spatial Sparsity Layer and Adaptive Sparsity Algorithm, which effectively boosts sparsity in intermediate representations while maintaining competitive performance, surpassing standard regularizers.

- **Practical Real-World Applications**: The research addresses meaningful problems, such as RNA modeling, and demonstrates the ability to capture significant signals even when assumptions are violated, showcasing the method's robustness and relevance in practical scenarios.

**Weaknesses**
- **Unclear Distinction Between Motifs and Noise**: The model's mechanism for differentiating motifs from noise is not explicitly explained, leading to ambiguity in its real-world application.

- **Lack of Architectural Details**: The neural architecture of the model is inadequately described, with only a sparse equation provided, leaving gaps in understanding its implementation and learning process.

- **Absence of Visualizations**: While some figures illustrate motifs, there is a lack of visualizations that demonstrate the model's ability to attend to or decode these motifs, limiting interpretability beyond reported accuracies.

- **Dependence on Strong Assumptions**: The theorem relies on stringent constraints like NON-OVERLAPPING motifs and PATCH-INDEPENDENCE, which may not hold in practical scenarios, raising concerns about the model's robustness and applicability to real-world tasks.

**Decision**

The paper is recommended for acceptance, as it offers a strong theoretical foundation with the Motif Identifiability Theorem, introduces an innovative algorithm for enhancing sparsity in deep learning, and demonstrates practical applications in RNA modeling, proving its robustness and relevance.

**Reviewer Concerns:**

The author's feedback effectively addressed most concerns, enhancing the overall quality of the paper. Additionally, they clarified several ambiguous points, leading most reviewers to consider a slight increase in their scores.
The review scored with 0 has been discarded since there are not enough arguments to argue the rating.

**Reviewer Scores:**

The author's feedback effectively addressed most concerns, enhancing the overall quality of the paper. Additionally, they clarified several ambiguous points, leading most reviewers to consider a slight increase in their scores.

---

### Decision · Program_Chairs · 2026-01-26

Accept (Poster)